# Disrupting HIV-1 capsid formation causes cGAS sensing of viral DNA

Rebecca P Sumner[*] [ID], Lauren Harrison, Emma Touizer [ID], Thomas P Peacock[†], Matthew Spencer, Lorena Zuliani-Alvarez & Greg J Towers

## Abstract

Detection of viral DNA by cyclic GMP-AMP synthase (cGAS) is a first line of defence leading to the production of type I interferon (IFN). As HIV-1 replication is not a strong inducer of IFN, we hypothesised that an intact capsid physically cloaks viral DNA from cGAS. To test this, we generated defective viral particles by treatment with HIV-1 protease inhibitors or by genetic manipulation of *gag*. These viruses had defective Gag cleavage, reduced infectivity and diminished capacity to saturate TRIM5α. Importantly, unlike wild-type HIV-1, infection with cleavage defective HIV-1 triggered an IFN response in THP-1 cells that was dependent on viral DNA and cGAS. An IFN response was also observed in primary human macrophages infected with cleavage defective viruses. Infection in the presence of the capsid destabilising small molecule PF-74 also induced a cGAS-dependent IFN response. These data demonstrate a protective role for capsid and suggest that antiviral activity of capsid- and protease-targeting antivirals may benefit from enhanced innate and adaptive immunity *in vivo*.

**Keywords** capsid; cGAS; DNA sensing; HIV-1; protease inhibitor
**Subject Categories** Immunology; Microbiology, Virology & Host Pathogen Interaction
**The EMBO Journal (2020) 39: e103958**

## Introduction

The innate immune system provides the first line of defence against invading pathogens such as viruses. Cells are armed with pattern recognition receptors (PRRs) that recognise pathogen-associated molecular patterns (PAMPs), such as viral nucleic acids, and lead to the activation of a potent antiviral response in the form of secreted interferons (IFNs), proinflammatory cytokines and chemokines, the expression of which is driven by the activation of key transcription factors such as IFN regulatory factor 3 (IRF3) and nuclear factor kappa-light-chain-enhancer of activated B cells (NF-κB) (Chow *et al*, 2015). For HIV-1, a number of cytosolic PRRs have been

demonstrated to contribute to the detection of the virus in infected cells including DNA sensors cyclic GMP-AMP synthase (cGAS) (Gao *et al*, 2013; Lahaye *et al*, 2013; Rasaiyaah *et al*, 2013), IFI16 (Jakobsen *et al*, 2013; Jonsson *et al*, 2017), PQBP1 (Yoh *et al*, 2015), RNA sensors DDX3 (Gringhuis *et al*, 2017) and also MDA5, although only in the circumstance where the genome lacked 2′-*O*-methylation by 2′-*O*-methyltransferase FTSJ3 (Ringeard *et al*, 2019). The nuclear protein NONO has also been implicated in the detection of HIV cDNA (Lahaye *et al*, 2018). The best studied HIV sensor is cGAS, which upon binding double-stranded DNA, such as HIV-1 reverse transcription (RT) products, produces second messenger 2′3′-cGAMP (Ablasser *et al*, 2013; Sun *et al*, 2013; Wu *et al*, 2013) that binds and induces phosphorylation of ER-resident adaptor protein STING and its translocation to perinuclear regions (Tanaka & Chen, 2012). Phosphorylation of STING provides a platform for the recruitment of TBK1 and IRF3 leading to IRF3 phosphorylation and its subsequent translocation to the nucleus to drive expression of IFN and IFN-stimulated genes (ISGs) (Liu *et al*, 2015; Shang *et al*, 2019). Activation of STING by 2′3′-cGAMP also activates IKK and the transcription of NF-κB-dependent genes (Ishikawa & Barber, 2008).

Of course, detection of infection by sensing is not universal and viruses are expected to hide their PAMPs and typically have mechanisms to antagonise specific sensors and downstream restriction factors. Work from our laboratory, and others, has demonstrated that primary monocyte-derived macrophages (MDMs) (Tsang *et al*, 2009; Rasaiyaah *et al*, 2013) and THP-1 cells (Cingoz & Goff, 2019) can be infected by wild-type (WT) HIV-1 without significant innate immune induction. However, MDM senses HIV-1 if, for example, mutations are made in the capsid protein to prevent the recruitment of cellular cofactors such as CPSF6 and cyclophilin A (Rasaiyaah *et al*, 2013), after depletion of the cellular exonuclease TREX1 (Yan *et al*, 2010; Rasaiyaah *et al*, 2013) and HIV can be sensed by a process requiring NONO if restriction by SAMHD1 is overcome (Lahaye *et al*, 2018). Sensing of HIV was found to be dependent on viral reverse transcription (RT) and the cellular DNA sensing machinery cGAS and STING. In addition to having a role in recruitment of cofactors for nuclear entry, a variety of evidence suggests that the viral capsid has a role in protecting the process of viral DNA synthesis, preventing degradation of RT products by cellular

Division of Infection and Immunity, University College London, London, UK
*Corresponding author. Tel: +44 20 3108 2422; E-mail: r.sumner@ucl.ac.uk
†Present address: Department of Medicine, Imperial College London, London, UK

nucleases such as TREX1 and from detection by DNA sensors (Burdick *et al*, 2017; Francis & Melikyan, 2018).

Here, we have tested the hypothesis that an intact viral capsid is crucial for evasion of innate immune sensing by disrupting the process of viral particle maturation, either biochemically, using protease inhibitors (PIs), or genetically, by mutating the cleavage site between the capsid protein and spacer peptide 1. The resulting viral particles had defective Gag cleavage, reduced infectivity and, unlike wild-type HIV-1, activated an IFN-dependent innate immune response in THP-1 cells and primary human macrophages. This response in THP-1 cells was mostly dependent on viral DNA synthesis and the cellular sensors cGAS and STING. Defective viruses were less able to saturate restriction by TRIM5α indicating a reduced ability to bind this restriction factor, likely due to aberrant particle formation. Finally, we show that the viral capsid-binding small molecule inhibitor PF-74, which has been proposed to accelerate capsid opening (Marquez *et al*, 2018), also induces HIV-1 to activate an innate response in THP-1 cells, which is dependent on cGAS. Together these data support the hypothesis that the viral capsid plays a physical role in protecting viral DNA from the cGAS/STING sensing machinery in macrophages and that disruption of Gag cleavage and particle maturation leads to aberrant viral capsid formation and activation of an IFN response that may be important *in vivo* during PI treatment of HIV-1.

## Results

### Protease inhibitor treatment of HIV-1 leads to innate immune induction in macrophages

To test the hypothesis that intact viral capsids protect HIV-1 DNA from detection by DNA sensors, we sought to activate sensing using defective viral particles with disrupted capsid maturation. The protease inhibitor (PI) class of anti-retrovirals blocks the enzymatic activity of the viral protease, preventing Gag cleavage and proper particle formation, as observed by electron microscopy (Schatzl *et al*, 1991; Muller *et al*, 2009). By producing VSV-G-pseudotyped HIV-1ΔEnv.GFP (LAI strain (Peden *et al*, 1991) with the Nef coding region replaced by GFP, herein called HIV-1 GFP) in the presence of increasing doses of the PI lopinavir (LPV, up to 100 nM), we were able to generate viral particles with partially defective Gag cleavage, as assessed by immunoblot of extracted viral particles detecting HIV-1 CA protein (Fig 1A). At the highest dose of LPV (100 nM), increased amounts of intermediate cleavage products corresponding to capsid and spacer peptide 1 (CA-SP1), matrix and CA (MA-CA), MA, CA, SP1 and nucleocapsid (MA-NC) were particularly evident along with increased amounts of full length uncleaved Gag (Figs 1A and EV1A). Uncleaved CA-SP1 was also evident at 30 nM LPV. As expected, defects in Gag cleavage were accompanied by a reduction in HIV-1 GFP infectivity in both phorbol myristyl acetate (PMA)-treated THP-1 (Fig 1B) and U87 cells (Fig 1C). For the highest dose of LPV, this corresponded to a 24- and 48-fold defect in infectivity in each cell type, respectively. Viral titres were calculated according to the number of genomes, assessed by qPCR (see Methods), to account for small differences in viral production between conditions. These differences were no more than twofold from untreated virus.

To test the visibility of PI-inhibited viruses to innate sensing responses, we generated a THP-1 cell line that was stably depleted for the HIV restriction factor SAMHD1 (Appendix Fig S1A). Monocytic THP-1 cells can be differentiated to a macrophage-like transcriptome by treatment with PMA, to yield an adherent cell line that is highly competent for innate immune sensing, including DNA sensing. Differentiation of THP-1 normally leads to SAMHD1 activation by dephosphorylation and potent restriction of HIV-1 infection (Cribier *et al*, 2013). SAMHD1 depletion effectively relieved this restriction and allowed HIV-1 GFP infection (Appendix Fig S1B). SAMHD1-depleted THP-1 cells (herein referred to as THP-1 shSAMHD1 cells) remained fully competent for innate immune sensing and produced interferon-stimulated genes (ISGs) and inflammatory chemokines including *CXCL-10*, *IFIT-2* (also known as *ISG54*) and *CXCL-2* in response to a range of stimuli, including transfection of herring testis DNA (HT-DNA), exposure to 2′3′-cGAMP and infection by Sendai virus (Appendix Fig S1C–E).

Infection of PMA-treated THP-1 shSAMHD1 cells with HIV-1 GFP that had been produced in the presence of increasing doses of LPV led to a virus and LPV dose-dependent increase in the expression of ISGs *CXCL-10*, *IFIT-2* and *MxA* at the mRNA level (Fig 1D–F), and CXCL-10 protein secretion (Fig 1G). In agreement with previous reports (Cingoz & Goff, 2019), HIV-1 GFP produced in the absence of LPV induced very little, or no ISG expression in THP-1 cells at the doses tested, consistent with the hypothesis that HIV-1 shields its PAMPs from cellular PRRs (see Fig 1D–G, 0 nM drug dose). Virus dose in these experiments was normalised according to RT activity, as measured by SG-PERT (see Methods), which differed no more than fivefold in the LPV-treated versus LPV-untreated virus. Determination of genome by qRT-PCR gave similar dose values. Infection levels in differentiated THP-1 cells were approximately equivalent between the various LPV doses tested (Fig EV1B) because HIV-1 GFP infection of THP-1 is maximal at about 70% GFP positivity (Pizzato *et al*, 2015). Similar results were obtained with the PI darunavir (DRV); treatment of HIV-1 GFP with increasing doses of DRV (up to 50 nM) led to defects in Gag cleavage (Fig EV1C), decreased infectivity (Fig EV1D and E) and at 12.5 and 25 nM DRV-treated virus activated an ISG response in PMA-treated THP-1 shSAMHD1 cells (Fig EV1F and G).

To test whether LPV-treated HIV-1-induced ISG expression depended on IFN production or direct activation of ISGs, infections were repeated in the presence of the JAK1/2 inhibitor ruxolitinib (Quintas-Cardama *et al*, 2010). Activation of STAT transcription factor downstream of IFN receptor engagement requires phosphorylation by JAKs, and hence, ruxolitinib inhibits IFN signalling (Fig 1H). Induction of *MxA* (Fig 1H) and *CXCL-10* (Fig EV1H) expression by LPV-treated HIV-1 GFP was severely reduced in the presence of ruxolitinib, indicating that induction of ISG expression in these experiments requires an infection-driven type I IFN response. Treatment of cells with type I IFN provided a positive control for ruxolitinib activity (Figs 1H and EV1H). Importantly, measurement of viral DNA production in infected PMA-treated THP-1 shSAMHD1 cells, demonstrated that LPV did not increase DNA levels, ruling out increased DNA levels as an explanation for increased sensing (Fig 1I). We conclude that PI-inhibited HIV-1 fails to protect viral DNA from innate immune sensors by effective encapsidation.

To test whether PI inhibition of HIV-1 caused similar innate immune activation in primary human cell infection, we turned to

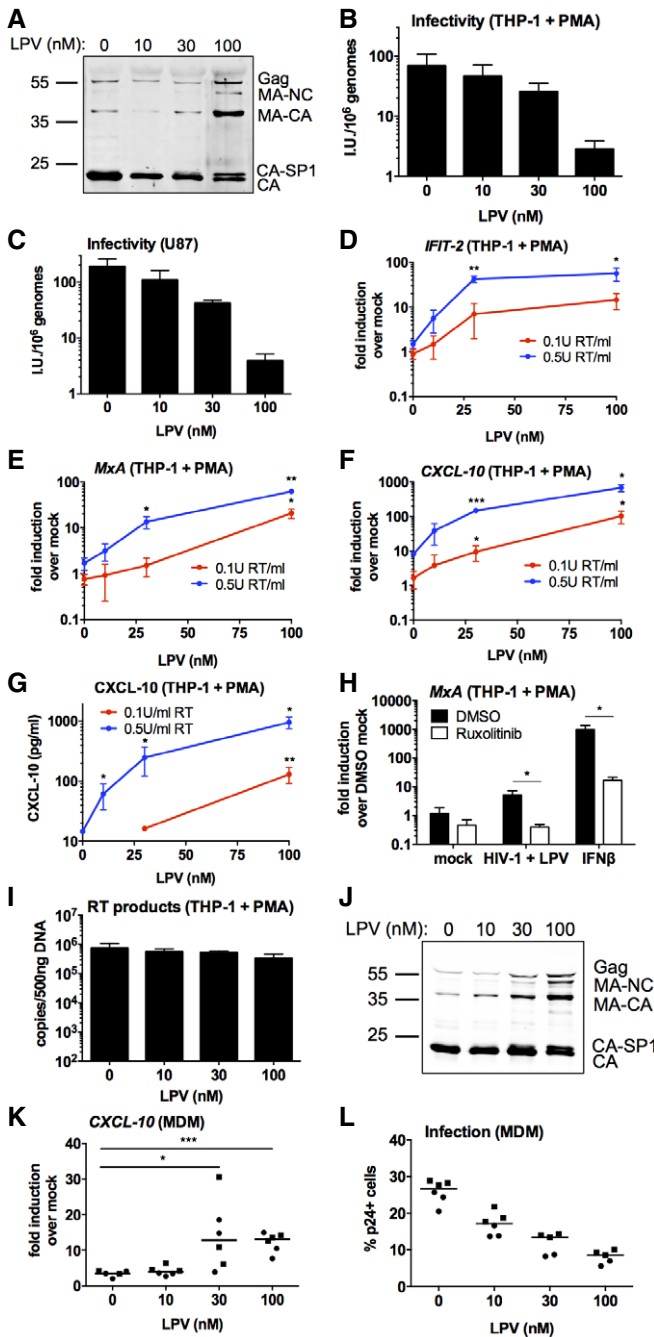

Figure 1.  PI treatment induces HIV-1 to trigger an ISG response in macrophages.

A      Immunoblot of HIV-1 GFP virus particles ($2 \times 10^{11}$ genomes) produced with lopinavir (LPV, 0-100 nM) detecting p24.

B, C   Titration of LPV-treated HIV-1 GFP viruses on PMA-treated THP-1 shSAMHD1 (B) or U87 (C) cells. Mean ± SD, $n = 3$.

D–F    ISG qPCR from PMA-treated THP-1 shSAMHD1 cells transduced for 24 h with LPV-treated HIV-1 GFP viruses (0.1 U RT/ml red line, 0.5 U RT/ml blue line).

G      CXCL-10 protein in cell supernatants from (D-F) (ELISA).

H      ISG qRT-PCR from PMA-treated THP-1 shSAMHD1 cells transduced for 24 h with 0.2 U RT/ml 30 nM LPV-treated HIV-1 GFP in the presence of DMSO vehicle or 2 µM ruxolitinib. A control was stimulated with 1 ng/ml IFNβ.

I      RT products from PMA-treated THP-1 shSAMHD1 cells transduced for 24 h with 0.3 U RT/ml LPV-treated HIV-1 GFP viruses.

J      Immunoblot of HIV-1 R9 BaL virus particles ($2 \times 10^{11}$ genomes) produced with LPV (0–100 nM) detecting p24.

K, L   ISG qRT-PCR (K) and infection data (L) from primary monocyte-derived macrophages (MDMs) infected for 24 h with LPV-treated HIV-1 R9 BaL viruses (0.2 U RT/ml). Data are collated from two donors (represented by circles and squares), $n = 3$. Horizontal line represents the median.

Data information: Data are mean ± SD, $n = 3$, representative of 2 repeats (H, I), or 3 repeats (D–G). Statistical analyses were performed using the Student's $t$-test, with Welch's correction where appropriate and comparing to the 0 nM LPV condition. $*P < 0.05$, $**P < 0.01$, $***P < 0.001$. See also Fig EV1.  For experiments in which the virus dose used was normalised by RT activity, the number of genome copies was also measured by qPCR of virus. This gave dose equivalents of within twofold to threefold of RT equivalents.

Source data are available online for this figure.

treated HIV-1 GFP induced more CXCL-10 secretion in primary MDM than untreated HIV-1 GFP (0 nM DRV) and this was dependent on type I IFN production, as evidenced by the lack of CXCL-10 production in the presence of ruxolitinib (Fig EV1K). Infection levels were not changed by ruxolitinib treatment (Fig EV1L). Together, these data suggest that infection by PI-treated HIV-1 induces an IFN-dependent innate immune response in PMA-treated THP-1 cells and primary human MDM that is not observed after infection with untreated virus.

## HIV-1 bearing Gag cleavage mutations also induces innate immune activation

Producing virus in the presence of PI suppresses Gag cleavage at multiple sites. Previous work suggested that inhibition of the CA-SP1 cleavage site was particularly toxic to infectivity and defective particles were irregular with partial polyhedral structures (Muller *et al*, 2009; Mattei *et al*, 2018). Concordantly, our data show a defect in cleavage at the CA-SP1 site in the presence of LPV (Fig 1A and J) or DRV (Fig EV1C). Importantly, the presence of even small proportions of CA-SP1 cleavage mutant exerted trans-dominant negative effects on HIV-1 particle maturation (Muller *et al*, 2009). To test whether a CA-SP1 cleavage defect can cause HIV-1 to trigger innate sensing, we prepared chimeric VSV-G pseudotyped HIV-1 GFP viruses by transfecting 293T cells with varying ratios of WT HIV-1 GFP and HIV-1 GFP with CA-SP1 Gag mutant L363I M367I (Wiegers *et al*, 1998; Checkley *et al*, 2010). Increasing the proportion of the ΔCA-SP1 mutant increased the presence of uncleaved CA-SP1 detected by immunoblot (Fig 2A). Defective cleavage was accompanied by a modest decrease in infectivity on U87 cells (Fig 2B).

HIV-1 R9 (BaL-Env) infection of primary human macrophages. Production of R9 (BaL-Env) in HEK293T cells in the presence of 10–100 nM LPV induced the expected defects in Gag cleavage (Fig 1J) and infectivity (Fig EV1I and J) as observed with VSV-G-pseudotyped HIV-1 GFP (Fig 1A–C). Furthermore, virus produced in the presence of 30 and 100 nM LPV induced the expression of *CXCL-10* on infection of primary MDM, whereas virus grown in the absence of LPV, or at low LPV concentrations (10 nM), induced very little *CXCL-10* expression (Fig 1K). Increasing concentrations of LPV during HIV-1 production led to a decrease in MDM infection, read out by p24 positivity, in these experiments (Fig 1L). Similarly, DRV-

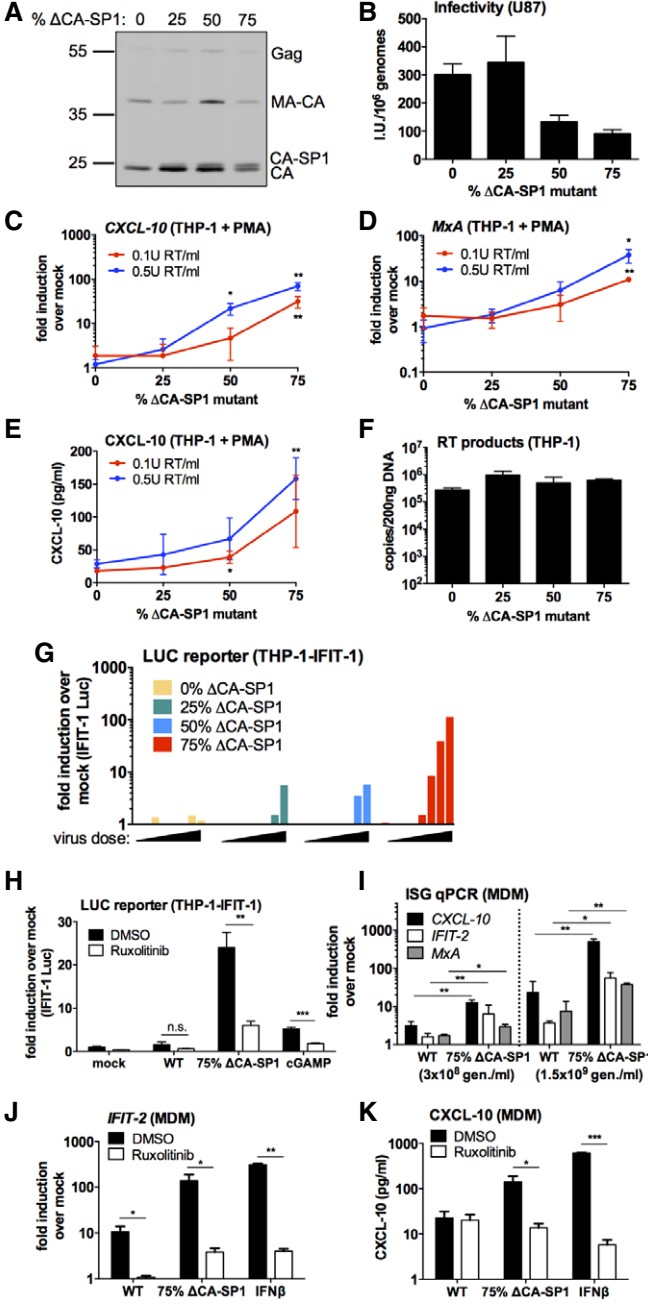

**Figure 2. HIV-1 with Gag protease cleavage mutation induces ISGs in macrophages.**

A    Immunoblot of HIV-1 GFP virus particles (2 × 10$^{11}$ genomes) with varying proportions of ΔCA-SP1 protease cleavage mutation detecting p24.

B    Titration of HIV-1 GFP ΔCA-SP1 viruses on U87 cells. Mean ± SD, n = 3.

C, D    ISG qRT–PCR from PMA-treated THP-1 shSAMHD1 cells transduced for 24 h with HIV-1 GFP ΔCA-SP1 viruses (0.1 U RT/ml red line, 0.5 U RT/ml blue line).

E    CXCL-10 protein in supernatants from (C, D) (ELISA).

F    RT products from THP-1 cells transduced for 24 h with 6 × 10$^9$ genomes/ml (approx. 0.5 U RT/ml) HIV-1 GFP ΔCA-SP1 viruses.

G    IFIT-1 reporter activity from monocytic THP-1-IFIT-1 cells transduced for 24 h with HIV-1 GFP ΔCA-SP1 viruses (0.016 − 0.2 U RT/ml). Data are shown as individual measurements, representative of 2 repeats.

H    IFIT-1 reporter activity from monocytic THP-1-IFIT-1 cells transduced with HIV-1 GFP containing either 0% (WT) or 75% ΔCA-SP1 mutant, or stimulated with 4 μg/ml cGAMP as a control, in the presence of DMSO vehicle or 2 μM ruxolitinib.

I    ISG qRT–PCR from primary MDM transduced for 24 h with WT HIV-1 GFP or 75% ΔCA-SP1 mutant (3 × 10$^8$ genomes/ml or 1.5 × 10$^9$ genomes/ml, equivalent to 0.1 U RT/ml and 0.5 U RT/ml).

J    ISG qRT–PCR from primary MDM transduced for 24 h with WT HIV-1 GFP or 75% ΔCA-SP1 mutant (1.5 × 10$^9$ genomes/ml), or stimulated with 1 ng/ml IFNβ, in the presence of DMSO vehicle or 2 μM ruxolitinib.

K    CXCL-10 protein in supernatants from (J) (ELISA).

Data information: Data are mean ± SD, n = 3, representative of 2 repeats (F, I-K) or 3 repeats (C-E, H). Statistical analyses were performed using the Student's t-test, with Welch's correction where appropriate and comparing to the 0% ΔCA-SP1 virus (C–E, I) or the DMSO control (H, J, K). *P < 0.05, **P < 0.01, ***P < 0.001. See also Fig EV2.

Source data are available online for this figure.

production rather than inhibition of RT activity by the ΔCA-SP1 mutation. Cleavage defective viruses, and not wild-type virus, also induced dose-dependent luciferase expression from an undifferentiated THP-1 cell line that had been modified to express Gaussia luciferase under the control of the *IFIT-1* (also known as *ISG56*) promoter, herein called IFIT1-luc (Mankan *et al*, 2014) (Fig 2G). IFIT1-luc is both IRF-3- and IFN-sensitive (Mankan *et al*, 2014). HIV-1 bearing ΔCA-SP1 mutant also induced a type I IFN response, evidenced by suppression of IFIT1-luc by ruxolitinib (Fig 2H). In the IFIT1-luc cells, ΔCA-SP1 mutation did not impact infection levels (Fig EV2A–C) and neither did ruxolitinib treatment (Fig EV2C). We propose that during single round infection of THP-1 cells, the virus has already integrated by the time IFN is produced, and this is why ruxolitinib does not rescue infection and thus the percentage of GFP-positive cells. To corroborate these findings in primary cells, we infected MDM with HIV-1 GFP ΔCA-SP1 (75% mutant) and found enhanced *CXCL-10, IFIT-2 and MxA* expression compared with WT HIV-1 GFP (Figs 2I and EV2D). Furthermore, HIV-1 GFP ΔCA-SP1 induced an IFN response in these cells, as treatment with ruxolitinib significantly reduced *IFIT-2* expression (Fig 2J) and CXCL-10 secretion (Fig 2K) induced by HIV-1 GFP ΔCA-SP1. Interestingly in primary MDM, treatment of cells with ruxolitinib did enhance infection levels of HIV-1 GFP ΔCA-SP1, but not WT HIV-1 GFP. This is consistent with the notion that HIV-1 GFP ΔCA-SP1 induces a IFN-dependent antiviral response in these cells that is, in this case, fast enough to inhibit single round infection (Fig EV2E and F).

We also performed similar experiments measuring replication of HIV-1 in MDM over several days, inhibiting replication with various

As with HIV-1 GFP produced in the presence of PIs, infection of PMA-treated THP-1 shSAMHD1 cells with the HIV-1 GFP ΔCA-SP1 mutants led to a ΔCA-SP1 dose-dependent increase in the expression of *CXCL-10* (Fig 2C) and *MxA* mRNA (Fig 2D), and CXCL-10 at the protein level (Fig 2E). Induction was not explained by differences in the amount of viral DNA in infected cells, and similar levels of viral DNA (Fig 2F) and infection (Fig EV2A) were observed at the viral doses tested. Virus dose in these experiments was normalised according to RT activity, which differed no more than fivefold between viruses. Importantly, differences in RT activity, measured by SG-PERT, were mirrored by measurements of genome copy, measured by qPCR. This is consistent with variation in viral

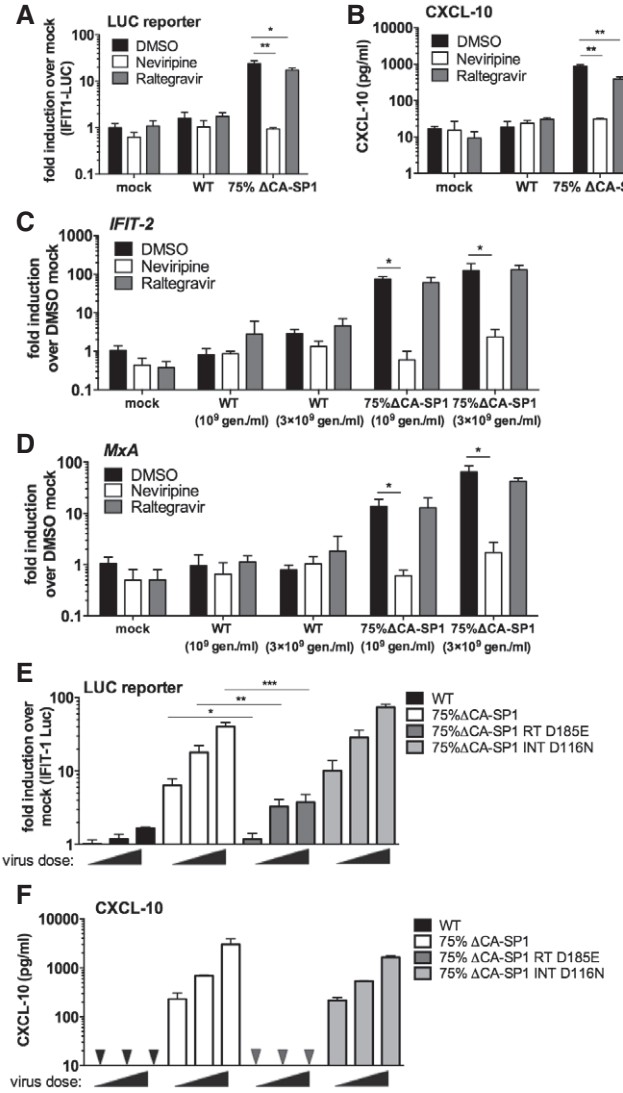

**Figure 3. Innate immune activation is RT-dependent.**

A    IFIT-1 reporter activity from THP-1-IFIT-1 cells transduced for 24 h with HIV-1 GFP containing 0% (WT) or 75% ΔCA-SP1 mutant (1 U RT/ml) in the presence of DMSO vehicle, 5 μM nevirapine or 10 μM raltegravir.

B    CXCL-10 protein in supernatants from (A) (ELISA).

C, D  ISG qRT-PCR from THP-1-IFIT-1 cells transduced for 24 h with 0% (WT) or 75% ΔCA-SP1 mutant ($10^9$ and $3 \times 10^9$ genomes/ml) in the presence of DMSO vehicle, 5 μM nevirapine or 10 μM raltegravir.

E    IFIT-1 reporter activity from THP-1-IFIT-1 cells transduced for 24 h with HIV-1 GFP containing 0% ΔCA-SP1 (WT), 75% ΔCA-SP1, 75% ΔCA-SP1 carrying a mutation in reverse transcriptase (75% ΔCA-SP1 RT D185E) or 75% ΔCA-SP1 carrying a mutation in integrase (75% ΔCA-SP1 INT D116N) ($3.75 \times 10^9$, $7.5 \times 10^9$ and $1.5 \times 10^{10}$ genomes/ml).

F    CXCL-10 protein in supernatants from (E) (ELISA). Triangles indicate CXCL-10 not detected.

Data information: Data are mean ± SD, *n* = 3, representative of 2 repeats. Statistical analyses were performed using Student's *t*-test, with Welch's correction where appropriate. *$P < 0.05$, **$P < 0.01$, ***$P < 0.001$. See also Fig EV3.

concentrations of LPV. In this case, neither blockade of IFN receptor with antibody, or inhibition of JAK/STAT signalling with ruxolitinib, significantly rescued infection over two independently

performed experiments (Appendix Fig S2A–D). We hypothesise that prevention of IFN activity does not rescue viral replication because the replication inevitably remains suppressed by effective protease inhibition. However, *in vivo,* we might expect that IFN produced in this way would contribute to innate and adaptive immune suppression of infection. Together these data support our hypothesis that disruption of Gag maturation yields viral particles that fail to shield PAMP from innate sensors.

### Maximal innate immune activation by maturation defective viruses is dependent on viral DNA synthesis

To determine whether viral DNA synthesis is required for HIV-1 bearing ΔCA-SP1 to trigger sensing, we infected THP-1 IFIT1-luc cells with HIV-1 75% ΔCA-SP1 in the presence of reverse transcriptase inhibitor nevirapine and assessed sensing by measuring IFIT1-luc expression and CXCL10 secretion. As expected, infectivity was severely diminished by 5 μM nevirapine (Fig EV3A and B) and both luciferase (Fig 3A) and CXCL-10 (Fig 3B) secretion was completely inhibited suggesting that viral DNA synthesis is required to activate sensing. Concordantly, expression of ISGs *IFIT-2* (Fig 3C) and *MxA* (Fig 3D) induced by HIV-1 75% ΔCA-SP1 was also abolished in the presence of nevirapine. A small, but statistically significant, reduction in luciferase (Fig 3A) and CXCL-10 (Fig 3B) secretion was observed in the presence of the integrase inhibitor raltegravir, although this was not observed in every experiment (Fig 3C and D). We conclude that viral DNA is the active PAMP and this notion was also supported by the observation that mutation D185E in the RT active site (HIV-1 ΔCA-SP1 RT D185E) also reduced activation of IFIT-1 luc expression (Fig 3E) and CXCL10 secretion (Fig 3F) on infection of the THP-1 IFIT-1 reporter cells. Mutation D116N of the viral integrase (HIV-1 ΔCA-SP1 INT D116N) impacted neither luciferase induction (Fig 3E) nor CXCL-10 (Fig 3F) secretion.

Surprisingly neither treatment with 10 μM raltegravir (Fig EV3A and B), or infection with HIV-1 ΔCA-SP1 INT D116N (Fig EV3C), led to a reduction in GFP positivity in monocytic THP-1 cells. Importantly, GFP expression was suppressed by raltegravir in parallel infection of PMA-treated THP-1 cells (Fig EV3D and E) confirming that integration was indeed inhibited by 10 μM raltegravir or D116N integrase mutation. We propose that the GFP positivity observed in monocytic THP-1 cells, in the presence of raltegravir, or by ΔCA-SP1 INT D116N, is due to expression from 2′-LTR circles as has been observed in other cell types (Van Loock *et al*, 2013; Bonczkowski *et al*, 2016).

### Viral DNA of maturation defective HIV-1 is sensed by cGAS and STING

To investigate which innate sensors were involved in detecting cleavage defective HIV-1, we infected cells that had been genetically manipulated by CRISPR/Cas 9 technology to lack the DNA sensing component proteins cGAS (Invivogen) or STING (Tie *et al*, 2018), or the RNA sensing component MAVS (Tie *et al*, 2018). As expected, STING$^{-/-}$ cells did not respond to transfected herring testis (HT) DNA but ISG induction was maintained in response to the RNA mimic poly I:C (Fig 4A). MAVS$^{-/-}$ cells showed the opposite phenotype, responding to poly I:C, but not HT-DNA (Fig 4A). As expected, Dual IRF reporter THP-1 cells, knocked out for cGAS (Invivogen), responded normally to poly I:C, LPS and cGAMP but not transfected HT-DNA

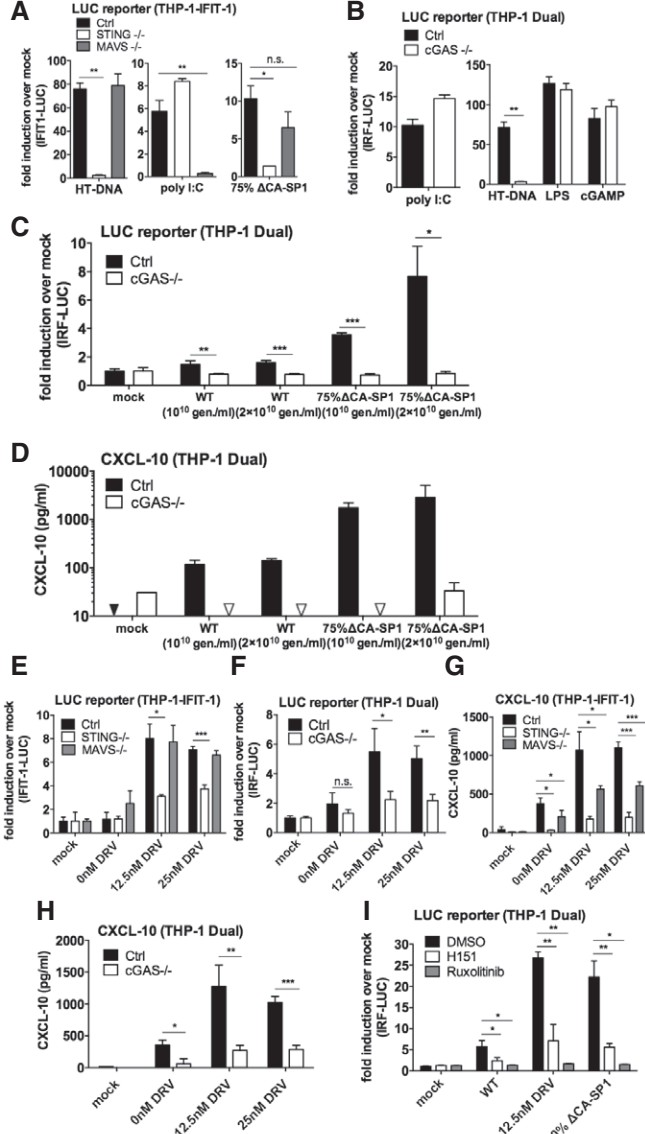

Figure 4. Innate immune activation is DNA sensing-dependent.
A    IFIT-1 reporter activity from PMA-treated THP-1-IFIT-1 shSAMHD1 cells lacking STING or MAVS, or a gRNA control (Ctrl) cell line transduced for 24 h with HIV-1 GFP 75% ΔCA-SP1 (0.4 U RT/ml) or stimulated by transfection with either HT-DNA (0.1 µg/ml) or poly I:C (0.5 µg/ml).
B, C  IRF reporter activity from PMA-treated THP-1 Dual shSAMHD1 cells lacking cGAS or a matching control (Ctrl) cell line stimulated for 24 h with poly I:C (transfection, 0.5 µg/ml), HT-DNA (transfection, 0.1 µg/ml), LPS (50 ng/ml) or cGAMP (transfection, 0.5 µg/ml) (B) or transduced for 24 h with HIV-1 GFP containing either 0% (WT) or 75% ΔCA-SP1 (1 × 10^10 and 2 × 10^10 genomes/ml) (C).
D    CXCL-10 protein in supernatants from (C) (ELISA). Triangles indicate CXCL-10 not detected.
E    IFIT-1 reporter activity from PMA-treated THP-1-IFIT-1 shSAMHD1 cells lacking STING, MAVS or matching gRNA control (Ctrl) cell line transduced for 24 h with DRV-treated HIV-1 GFP (1 × 10^10 genomes/ml).
F    IRF reporter activity from PMA-treated THP-1 Dual shSAMHD1 cells lacking cGAS or matching control (Ctrl) cell line transduced for 24 h with DRV-treated HIV-1 GFP (1 × 10^10 genomes/ml).
G    CXCL-10 protein in supernatants from (E) (ELISA).
H    CXCL-10 protein in supernatants from (F) (ELISA).
I    IRF reporter activity from PMA-treated THP-1 Dual shSAMHD1 control cells transduced for 48 h with WT, DRV-treated (DRV, 12.5 nM) or HIV-1 GFP containing 90% ΔCA-SP1 (1 × 10^10 genomes/ml) in the presence of DMSO vehicle, 2 µM ruxolitinib or 0.5 µg/ml H151.

Data information: Data are mean ± SD, n = 3, representative of 2 (E-I) or 3 (A-D) repeats. Statistical analyses were performed using Student's t-test, with Welch's correction where appropriate. *P < 0.05, **P < 0.01, ***P < 0.001. See also Fig EV4.

To corroborate data obtained in the CRISPR cell lines, infection assays were also repeated in THP-1 Dual reporter cells in the presence of the recently available STING inhibitor H151 (Haag *et al*, 2018). ISG induction by 12.5 nM DRV-treated or HIV-1 GFP bearing 90% ΔCA-SP1 was greatly reduced by the presence of H151 (Fig 4I), further supporting a role for DNA sensing in the detection of maturation defective HIV-1. As expected, IRF reporter activity was also suppressed by ruxolitinib (Fig 4I). Neither H151 nor ruxolitinib affected infection levels in these experiments (Fig EV4E).

## Maturation defective viruses fail to saturate TRIM5α in an abrogation-of-restriction assay

If maturation defective viruses consist of defective particles that have a reduced ability to protect viral DNA from cGAS, we hypothesised that these particles may also have a reduced capacity to bind the restriction factor TRIM5α. Rhesus monkey TRIM5α binds HIV-1 viral capsid and forms hexameric cage-like structures around the intact HIV capsid lattice (Ganser-Pornillos *et al*, 2011; Li *et al*, 2016). TRIM5α binding to viral capsid leads to proteasome recruitment, disassembly of the virus and activation of an innate response (Pertel *et al*, 2011; Fletcher *et al*, 2015, 2018). Viral restriction can be overcome by co-infection with high doses of a saturating virus in an abrogation-of-restriction assay, and this has been suggested to be dependent on the stability of the incoming viral capsid (Shi & Aiken, 2006; Jacques *et al*, 2016).

As a measure of HIV-1 core integrity, we tested the ability of the maturation defective viruses to saturate restriction by rhesus macaque TRIM5α. Rhesus FRhK4 cells were co-infected with a fixed dose of

(Fig 4B). Induction of IFIT1-luc activity in PMA-treated IFIT1-luc shSAMHD1 THP-1 cells by HIV-1 GFP bearing 75% ΔCA-SP1 was completely absent in STING knockout cells, but maintained in the MAVS knockout cells, consistent with DNA being the predominant viral PAMP detected (Fig 4A). Confirming these findings, no IRF reporter activity (Fig 4C) or CXCL-10 production (Fig 4D) was observed in PMA-treated THP-1 Dual shSAMHD1 cGAS$^{-/-}$ cells infected with HIV-1 75% ΔCA-SP1. Similar findings were also observed for DRV-treated wild-type HIV-1 GFP, where induction of IFIT1-luc reporter activity was dependent on STING (Fig 4E) and cGAS (Fig 4F), but not MAVS expression (Fig 4E). Interestingly, whilst CXCL-10 production in these experiments was severely diminished in STING$^{-/-}$ (Fig 4G) and cGAS$^{-/-}$ (Fig 4H) cells, levels were also reduced in MAVS$^{-/-}$ cells (Fig 4G) suggesting a contribution by HIV-1 RNA sensing in the production of this inflammatory cytokine. In all experiments, no significant difference in infection levels between the control and knockout cell lines was observed (Fig EV4A–D).

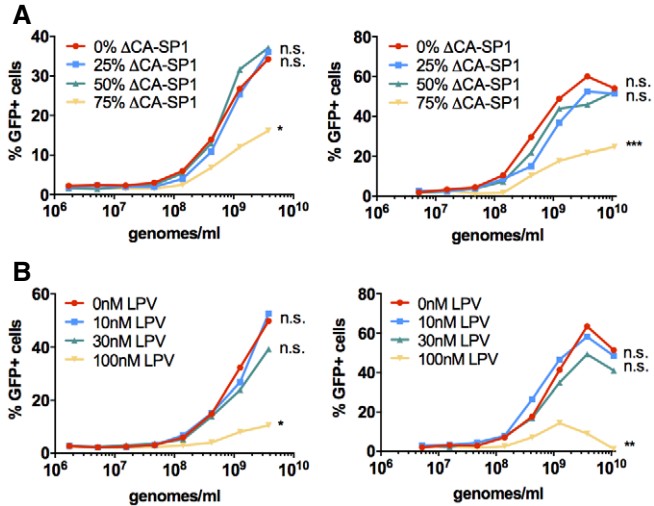

**Figure 5. Gag-defective HIV-1 particles are less able to saturate restriction factor TRIM5.**

A, B Abrogation-of-restriction assay in FRhK4 cells expressing restrictive rhesus TRIM5. FRhK4 cells were co-transduced with a fixed dose of HIV-1 GFP ($5 \times 10^7$ genomes/ml) and increasing doses of HIV-LUC ΔCA-SP1 mutants (A) or LPV-treated HIV-LUC viruses (B) ($1.7 \times 10^6$–$3.8 \times 10^9$ genomes/ml). Rescue of GFP infectivity was assessed by flow cytometry. Data are singlet % GFP values, and two repeats of the experiment are shown. See also Appendix Fig S3. Statistical analyses were performed using 2-way ANOVA with multiple comparisons. *$P < 0.05$, **$P < 0.01$, ***$P < 0.001$.

HIV-1 GFP and increasing doses of either wild-type untreated HIV-1 luc, LPV-treated HIV-1 luc or HIV-1 luc bearing ΔCA-SP1. Rescue of HIV-1 GFP infectivity from TRIM5α was assessed by flow cytometry measuring GFP-positive cells. Viruses that induced a strong innate response, i.e. virus bearing 75% ΔCA-SP1 mutant (Fig 5A, Appendix Fig S3A) or wild-type HIV-1 treated with 30 or 100 nM LPV (Fig 5B, Appendix Fig S3B) showed a reduced ability or failed to saturate TRIM5α restriction. These data are consistent with cleavage defective HIV-1 particles failing to form the authentic hexameric lattice required for recruitment of TRIM5α (Li *et al*, 2016; Ganser-Pornillos & Pornillos, 2019) and protection of genome.

**Treatment with viral capsid-binding small molecule PF-74 causes HIV-1 to trigger a DNA sensing-dependent ISG response**

Recent single molecule analysis of viral capsid uncoating demonstrated that the capsid-binding small molecule inhibitor of HIV, PF-74, accelerates capsid opening (Marquez *et al*, 2018). We therefore hypothesised that PF-74-treated HIV-1 may activate a DNA sensing-dependent innate immune response. To test this, we infected monocytic THP-1 IFIT-1 reporter cells with increasing doses of HIV-1 GFP (0.1–3 U/ml RT) in the presence or absence of 10 μM PF-74. This dose was sufficient to inhibit infection up to 1 U/ml RT HIV-1 GFP, indicating PF-74 is an effective inhibitor of HIV-1, although its potency could be improved (Fig 6A). Consistent with our hypothesis, at high-dose HIV-1 infection (3 U/ml RT), luciferase reporter induction was observed in the presence of PF-74 but not in the DMSO control (Fig 6B). ISG induction in the presence of 10 μM PF-74 was further confirmed in a second experiment by measuring

endogenous *CXCL-10* (Fig 6C) and *MxA* (Fig 6D) mRNA expression by qPCR and secreted CXCL-10 by ELISA in the IFIT1-luc reporter cells (Fig 6E). PF-74 treatment of HIV-1 GFP was further shown to induce a type I IFN response in these cells as IFIT1-luc reporter activity was diminished in the presence of ruxolitinib (Fig 6F). As expected, there was partial inhibition of infection with PF-74 and no significant difference in infection levels in the presence of ruxolitinib (Fig EV5A). Finally, we were able to demonstrate that innate sensing of PF-74-treated HIV-1 was dependent on cGAS as luciferase secretion by PF-74-treated HIV-1 GFP was lost in cGAS$^{-/-}$ cells (Fig 6G), but maintained in MAVS$^{-/-}$ cells (Fig 6H). As previously observed, the loss of cGAS (Fig EV5B) or MAVS (Fig EV5C) had no impact on HIV-1 infectivity suggesting sensing does not contribute to the inhibitory effect of PF74 in these single round infections. These results in THP-1 differ from our previous observation in MDM (Rasaiyaah *et al*, 2013), in which PF-74 treatment did not induce sensing of HIV-1. We assume that the 10 μM PF-74 used in MDM inhibited viral DNA (PAMP) synthesis, preventing cGAS activation.

## Discussion

Effective evasion of innate immune responses is expected to be crucial for successful infection, and all viruses have evolved countermeasures to hide PAMPs and/or directly reduce activation of the IFN response (Schulz & Mossman, 2016). Given the small coding capacity of HIV-1 and the general lack of innate activation observed with this virus *in vitro* (Lahaye *et al*, 2013; Rasaiyaah *et al*, 2013; Cingoz & Goff, 2019), we had hypothesised that HIV-1 uses its viral capsid to physically protect nucleic acid PAMPs from innate sensors such as cGAS. In this study, we used three approaches to demonstrate that the HIV-1 viral capsid plays a protective role in preventing IFN induction by viral DNA. By treating HIV-1 with PIs LPV (Fig 1) or DRV (Fig EV1), or mutating the cleavage site between CA and SP1 (Fig 2) we were able to generate aberrant particles by interfering with capsid maturation. In all cases, the resulting viruses had perturbations in Gag cleavage, reduced infectivity (Figs 1 and 2, and EV1) and had reduced capacity to saturate the restriction factor TRIM5α in an abrogation-of-restriction assay, indicative of altered stability/viral capsid integrity (Fig 5). Importantly, when these viruses were used to infect macrophages they induced a potent IFN response that was not observed on infection with untreated or WT HIV-1 (Figs 1 and 2, and EV1). Innate immune responses were almost entirely dependent on viral reverse transcription (Fig 3) and the cellular DNA sensing machinery comprising cGAS and STING (Fig 4), consistent with viral DNA being the most important PAMP in these experiments. As a third approach, our results were corroborated using the viral capsid targeting small molecule inhibitor PF-74, which has been proposed to accelerate capsid opening (Marquez *et al*, 2018). Treatment of HIV-1 with PF-74 also caused a DNA sensing-dependent IFN response (Fig 6). Together these data support a model in which the WT HIV-1 core remains intact as it traverses the cytoplasm, thus protecting viral DNA from detection by cGAS. Conversely, disruption of capsid maturation or integrity, either chemically or genetically, yields particles that fail to conceal viral DNA and thus activate a cGAS-dependent type 1 IFN response (Appendix Fig S4).

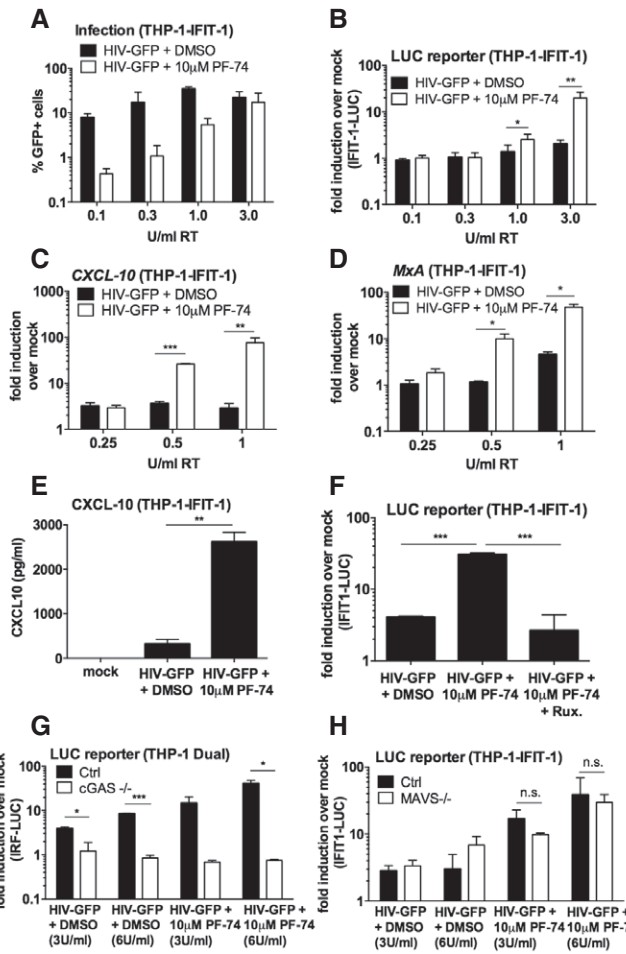

**Figure 6. PF-74 treatment induces HIV-1 to trigger a DNA sensing-dependent ISG response.**

A   Infection data for THP-1-IFIT-1 cells transduced for 24 h with HIV-1 GFP (0.1–3 U/ml RT) in the presence of DMSO vehicle or PF-74 (10 μM).

B   IFIT-1 reporter activity in supernatant from (A).

C, D   ISG qRT-PCR from monocytic THP-1-IFIT-1 cells transduced for 24 h with HIV-1 GFP (0.25–1 U/ml RT) in the presence of DMSO vehicle or PF-74 (10 μM).

E   CXCL-10 protein in supernatants of THP-1-IFIT-1 cells transduced for 24 h with HIV-1 GFP (3 U/ml) in the presence of DMSO vehicle or PF-74 (10 μM).

F   IFIT-1 reporter activity from THP-1-IFIT-1 cells transduced for 24 h with HIV-1 GFP (3 U/ml RT) in the presence of DMSO vehicle or PF-74 (10 μM) and ruxolitinib (Rux, 2 μM) as indicated.

G   IRF reporter activity from THP-1 Dual shSAMHD1 cells lacking cGAS or a matching control (Ctrl) cell line transduced for 24 h with HIV-1 GFP (3 and 6 U/ml) in the presence of DMSO vehicle or PF-74 (10 μM).

H   IFIT-1 reporter activity from THP-1-IFIT-1 cells lacking MAVS or a matching gRNA control (Ctrl) cell line transduced for 24 h with HIV-1 GFP (3 and 6 U/ml) in the presence of DMSO vehicle or PF-74 (10 μM).

Data information: Data are mean ± SD, n = 2 (C-E, G), 3 (A, F), 4 (H) or 6 (B), representative of 3 repeats. Statistical analyses were performed using Student's t-test, with Welch's correction where appropriate. *P < 0.05, **P < 0.01, ***P < 0.001. See also Fig EV5.

Previous studies have implicated a role for the HIV-1 viral capsid in evading an innate immune response using specific capsid mutants such as N74D and P90A, which prevent the recruitment of

cellular factors CPSF6 and cyclophilin A/Nup358, respectively (Lahaye et al, 2013; Rasaiyaah et al, 2013). More recent work has demonstrated that these mutant viruses are susceptible to host restriction factors that are normally evaded by the WT virus. For the P90A mutant, failure to recruit cyclophilin A makes the virus susceptible to restriction by TRIM5α (Kim et al, 2019), whilst the N74D mutant becomes susceptible to TRIM34-mediated restriction in a TRIM5α-dependent manner (Ohainle et al, 2020). Importantly, TRIM5α, and possibly TRIM34, can activate innate immune signalling pathways, likely explaining IFN induction when viruses engage these proteins in MDM (Pertel et al, 2011; Rasaiyaah et al, 2013; Uchil et al, 2013). The work presented herein suggests that, regardless of cofactor and restriction factor interactions, the HIV-1 viral capsid also plays a protective role and physically protects viral DNA from the sensor cGAS through a process of encapsidated DNA synthesis, which is disturbed by disturbing Gag cleavage. These findings are supported by a recent study demonstrating that HIV-1 bearing a PF-74-resistant, hyperstable capsid mutation (R143A) showed reduced cGAS-dependent innate induction, again linking capsid stability to sensing (Siddiqui et al, 2019).

We propose that HIV-1 has evolved to cloak viral DNA synthesis within an intact capsid (Rasaiyaah et al, 2013; Jacques et al, 2016). However, several studies have reported innate immune activation by WT HIV-1 in macrophages or dendritic cells. There are various reasons why study results may differ. For example, some studies have co-infected the cells with SIV VLPs bearing Vpx (Manel et al, 2010; Gao et al, 2013; Yoh et al, 2015; Johnson et al, 2018), which is known to degrade cellular proteins including SAMHD1, and the epigenetic regulator complex HUSH, and likely manipulates innate responses in complex ways. Other studies have used unpurified viral stocks (Manel et al, 2010; Yan et al, 2010), which may contain producer cell proteins, including cytokines, that can activate or stimulate target cell sensing capacity. The method by which primary cells are purified and differentiated may also affect their activation status and hence sensing capacity. For example Decalf et al describe activation of an IFN response in primary macrophages that was not dependent on reverse transcription or genomic RNA, attributing this response instead to detection of viral entry, dependent on the presence of a viral envelope (Decalf et al, 2017). In this study, MDM was prepared by positive selection using CD14[+] beads, which may activate the cells in ways that other purification methods do not. Another technical complication in testing whether HIV-1, or any other virus, triggers innate immune sensing is controlling for viral dose effects. We and others have found that at very high dose, HIV-1 activates innate immune pathways and this is particularly influenced by whether the viral supernatant is purified. In the experiments presented herein, all viruses were DNase treated and purified by centrifugation through sucrose and experiments were designed to control dose between variables. For example, viral dose was normalised by measuring RT activity (SG-PERT) or the number of viral genomes (qPCR) in viral preparations to account for differences in virus production, see legends. Critically, mutating the CA-SP1 cleavage site does not impact RT activity and treatment with protease inhibitors only inhibited supernatant RT activity at the highest dose used, and only by a few fold. We propose that small ISG responses to the doses of WT HIV-1 used here are likely due to low frequency detection of uncoating events because the minimal ISG response to WT HIV-1 was also dependent on cGAS (Figs 4C and 6G).

Further support for the important role of the viral capsid in innate immune evasion comes from data in dendritic cells demonstrating that unlike wild-type HIV-1, wild-type HIV-2 activates a strong RT- and cGAS-dependent IFN response. This difference in innate activation mapped to the viral capsid (Lahaye *et al*, 2013). Why the HIV-2 viral capsid, unlike the capsid of HIV-1, fails to protect RT products from innate sensors is the subject of ongoing investigation, but given that HIV-2 does not replicate in dendritic cells and macrophages (Duvall *et al*, 2007; Chauveau *et al*, 2015) these observations suggest that evasion of sensing by cGAS is a necessary requirement for replication in myeloid cells. Recent work has suggested that detection of HIV-2 viral cDNA by cGAS can occur in the nucleus of infected dendritic cells and macrophages, with a role for interaction of the nuclear protein NONO and the viral capsid (Lahaye *et al*, 2018). Concordantly, recent studies suggest that HIV-1 uncoats in the nucleus, rather than as previously thought, in the cytoplasm (Burdick *et al*, 2017, 2020; Francis & Melikyan, 2018; Bejarano *et al*, 2019). Here, we propose that PI-inhibited HIV-1 is detected in the cytoplasm because defective viral capsids are expected to fail in the cytoplasm, before they reach the nucleus, but this has not formally been proven. Further work will address whether nuclear HIV DNA can be detected by cGAS and how nuclear cGAS avoids detecting cellular DNA.

Interestingly, we discovered here that IFN triggering by single round infection of THP-1 cells did not lead to reduction in viral infectivity. This was particularly apparent in experiments using the JAK1/2 inhibitor ruxolitinib, which potently reduced the ISG response to viruses with defective viral capsids, but had no impact on infection read out by GFP positivity of the cells (Figs EV2C and EV4E, and EV5A). Similarly, cGAS knockout severely blunted ISG responses, but did not lead to a corresponding increase in GFP positivity (Figs EV4B and D, and EV5B). We propose that during single round infection of THP-1 cells the virus has already integrated by the time IFN is produced and GFP expression is not particularly sensitive to its antiviral effects. Indeed, the IFITM proteins (OhAinle *et al*, 2018; Petrillo *et al*, 2018; Yu & Liu, 2018), TRIM5α (Pertel *et al*, 2011; OhAinle *et al*, 2018) and MxB (Goujon *et al*, 2013; Kane *et al*, 2013; OhAinle *et al*, 2018) are the major IFN-induced inhibitors of HIV-1 in THP-1 cells and are not expected to impact GFP expression. There was also no rescue of LPV or DRV inhibited HIV-1 replication in primary MDM with either ruxolitinib or IFN receptor blockade with antibody (Fig EV1L, Appendix Fig S2). Conversely, infection by the 75% ΔCA-SP1 HIV-1 mutant in primary MDM was rescued by ruxolitinib (Fig EV2E and F). Thus, this virus, made by mixing WT and mutant Gag constructs, causes IFN production that subsequently inhibits viral infection. We expect that primary MDM may exhibit a faster or more potent IFN response allowing IFN-mediated suppression of infection even during single round infection of these cells. This result is reminiscent of rescue of infectivity, with IFN receptor blockade, of HIV-1 bearing CA mutants P90A and N74D in MDM (Rasaiyaah *et al*, 2013). We hypothesise that whether IFN inhibition rescues infection or not depends on the degree to which the suppression of replication depends on IFN secretion. For example, IFN inhibition does not rescue replication in PI-treated MDM because protease inhibition is sufficient for the observed viral inhibition. Conversely, in the case of Gag cleavage mutants (Fig EV2E and F), single round infection is in part inhibited by IFN and, thus, JAK/STAT inhibition rescues some degree of

infectivity. *In vivo*, whether the PI inhibited virus is directly suppressed by IFN or not, we would expect IFN to contribute to innate and adaptive immune responses and globally contribute to viral inhibition.

In this study, we have focused on infection of macrophages and macrophage-like THP-1 cells. Unfortunately, HIV does not replicate in primary human T cells, unless they are activated, typically by cross-linking and activating the T-cell receptor (TCR). This causes massive TCR-mediated signalling which is likely to overwhelm T-cell innate immune signalling driven by infection. The study of signalling induced by HIV therefore awaits the development of tractable T-cell infection models that do not require TCR activation.

An interesting finding that warrants further investigation is the observation that MAVS contributed to CXCL-10 production in response to infection with DRV-treated virus (Fig 4G), but did not contribute to the corresponding IFIT-1 reporter activity (Fig 4E). MAVS-dependent pathways are known to activate transcription factors other than IRF-3, such as NF-κB (Seth *et al*, 2005), which also contributes to the production of CXCL-10 (Yeruva *et al*, 2008), but not activation of the IFIT-1 reporter (Grandvaux *et al*, 2002). It is therefore possible that activation of MAVS by HIV-1 contributes to NF-κB activation in these cells but not an IRF-3 response.

In summary, these findings highlight the crucial role of the HIV-1 viral capsid in masking viral nucleic acids from innate immune sensors, particularly in protecting viral DNA from detection by cGAS/STING. As such, disrupting capsid integrity through mutation, treatment with protease inhibitors, or the capsid targeting small molecule PF-74 yields viral particles that fail to shield their PAMPs and thus activate a potent IFN response that is not observed with the WT virus. Together these data suggest that the therapeutic activity of viral capsid- or protease -targeting therapeutics, for example the recently described HIV-1 capsid inhibitor from Gilead Sciences (Yant *et al*, 2019), may be enhanced by induction of local antiviral IFN responses *in vivo* that could contribute to viral clearance by the innate and adaptive immune system. Furthermore, these findings encourage the design of therapeutics targeting viral capsids or structural proteins generally, which may also benefit from unmasking viral PAMPs and induction of innate immune responses.

## Materials and Methods

### Cells and reagents

HEK293T and U87 cells were maintained in DMEM (Gibco) supplemented with 10% foetal bovine serum (FBS, LabTech) and 100 U/ml penicillin plus 100 μg/ml streptomycin (Pen/Strep; Gibco). THP-1 cells were maintained in RPMI (Gibco) supplemented with 10% FBS and Pen/Strep. THP-1-IFIT-1 cells that had been modified to express Gaussia luciferase under the control of the *IFIT-1* promoter were described previously (Mankan *et al*, 2014). THP-1 Dual Control and cGAS$^{-/-}$ cells were obtained from Invivogen. Lopinavir (LPV), darunavir (DRV), nevirapine (NVP) and raltegravir were obtained from AIDS reagents. STING inhibitor H151 was obtained from Invivogen. JAK inhibitor ruxolitinib was obtained from CELL guidance systems. PF-74 was obtained from Sigma.

Lipopolysaccharide, IFNβ and poly I:C were obtained from Pepro-Tech. Sendai virus was obtained from Charles River Laboratories. Herring testis DNA was obtained from Sigma. cGAMP was obtained from Invivogen. Anti-IFNα/β receptor and control IgG2A antibodies were obtained from PBL Interferon Source and R&D systems, respectively. For stimulation of cells by transfection, transfection mixes were prepared using lipofectamine 2000 according to the manufacturer's instructions (Invitrogen).

### Generation of ΔCA-SP1, RT D185E and INT D116N viruses

pLAIΔEnv GFP/Luc ΔCA-SP1 (Gag mutant L363I M367I) was generated by two rounds of site-directed mutagenesis (using Pfu Turbo DNA polymerase, Agilent) using primers:

LAI_Gag_L363I fwd: 5′ CCGGCCATAAGGCAAGAGTTATCGCTGAA GCAATG 3′
LAI_Gag_L363I rev: 5′ GTTACTTGGCTCATTGCTTCAGCGATAACT CTTGC 3′
LAI_Gag_M367I fwd: 5′ GCAAGAGTTATCGCTGAAGCAATCAGC-CAAGTAAC 3′
LAI_Gag_M367I rev: 5′ GTAGCTGAATTTGTTACTTGGCTGATTGCT TCAGC 3′

pLAIΔEnv GFP and pLAIΔEnv GFP ΔCA-SP1 RT D185E and INT D116N were generated by site-directed mutagenesis using the following primers:

LAI_ RT D185E fwd: 5′ ATAGTTATCTATCAATACATGGAAGATTT GTATG 3′
LAI_ RT D185E rev: 5′ AAGTCAGATCCTACATACAAATCTTCCATG TATTG 3′
LAI_ INT D116N fwd: 5′ GGCCAGTAAAAACAATACATACAAACAA TGGCAGC 3′
LAI_ INT D116N rev: 5′ ACTGGTGAAATTGCTGCCATTGTTTGTAT GTATTG 3′

In all cases, mutated sequences were confirmed by sequencing, excised by restriction digestion and cloned back into the original plasmid.

### Isolation of primary monocyte-derived macrophages

Primary monocyte-derived macrophages (MDMs) were prepared from fresh blood from healthy volunteers. The study was approved by the joint University College London/University College London Hospitals NHS Trust Human Research Ethics Committee, and written informed consent was obtained from all participants. Peripheral blood mononuclear cells (PBMCs) were isolated by density gradient centrifugation using Lymphoprep (Stemcell Technologies). PBMCs were washed three times with PBS and plated to select for adherent cells. Non-adherent cells were washed away after 1.5 h and the remaining cells incubated in RPMI (Gibco) supplemented with 10% heat-inactivated pooled human serum (Sigma) and 40 ng/ml macrophage colony-stimulating factor (R&D systems). Cells were further washed after 3 days and the medium changed to RPMI supplemented with 10% heat-inactivated FBS. MDM was then infected 3–4 days later. Replicate experiments were performed with cells derived from different donors.

### Editing of cells by CRISPR/Cas 9

THP-1 IFIT-1 shSAMHD1 STING$^{-/-}$ and MAVS$^{-/-}$ cells were previously described (Tie *et al*, 2018). Briefly, lentiparticles to generate CRISPR/Cas9-edited cell lines were produced by transfecting 10-cm dishes of HEK293T cells with 1.5 µg of plentiCRISPRv2 encoding gene-specific guide RNAs (Addgene plasmid #52961), 1 µg of p8.91 packaging plasmid (Zufferey *et al*, 1997) and 1 µg of vesicular stomatitis virus-G glycoprotein expressing plasmid pMDG (Genscript) using Fugene 6 transfection reagent (Promega) according to the manufacturer's instructions. Virus supernatants were harvested at 48 and 72 h post-transfection, pooled and used to transduce THP-1 IFIT-1 shSAMHD1 cells by spinoculation (1,000 × *g*, 1 h, room temperature). Transduced cells were selected using puromycin (1 µg/ml, Merck Millipore) and single clones isolated by limiting dilution in 96-well plates. Clones were screened for successful gene knockout by luciferase assay and immunoblotting.

gRNA sequences:

STING: TCCATCCATCCCGTGTCCCAGGG
MAVS: CAGGGAACCGGGACACCCTC
Non-targeting control: ACGGAGGCTAAGCGTCGCAA

### Production of virus in 293T cells

HIV-1 and lentiviral particles were produced by transfection of HEK293T cells in T150 flasks using Fugene 6 transfection reagent (Promega) according to the manufacturer's instructions. For full length, HIV-1 with a BaL envelope cells was transfected with 8.75 µg pR9.BaL per flask. For HIV-1 GFP/Luc, each flask was transfected with 2.5 µg of vesicular stomatitis virus-G glycoprotein expressing plasmid pMDG (Genscript) and 6.25 µg pLAIΔEnv GFP/Luc. Virus supernatants were harvested at 48 and 72 h post-transfection, pooled, DNase treated (2 h at 37°C, DNaseI, Sigma) and subjected to ultracentrifugation over a 20% sucrose cushion. Viral particles were finally resuspended in RPMI supplemented with 10% FBS. For production of viruses in the presence of lopinavir or darunavir, the inhibitors were added at 24 h post-transfection and replaced after harvest at 48 h. Lentiparticles for SAMHD1 depletion were generated as previously described (Georgana *et al*, 2018). Viruses were titrated by infecting U87 cells ($10^5$ cells/ml) or PMA-treated THP-1 cells ($2 \times 10^5$ cells/ml) with dilutions of sucrose purified virus in the presence of polybrene (8 µg/ml, Sigma) for 48 h and enumerating GFP-positive cells by flow cytometry using the FACS Calibur (BD) and analysing with FlowJo software.

### SG-PERT

Reverse transcriptase activity of virus preparations was quantified by qPCR using a SYBR Green-based product-enhanced RT (SG-PERT) assay as described (Vermeire *et al*, 2012).

### Genome copy/RT products measurements

For viral genome copy measurements, RNA was extracted from 2 µl sucrose-purified virus using the RNeasy mini kit (Qiagen). The RNA was then treated with TURBO DNase (Thermo Fisher Scientific) and subjected to reverse transcription using Superscript III reverse transcriptase and random hexamers according to the manufacturer's

protocol (Invitrogen). Genome copies were then measured by TaqMan qPCR (Towers *et al*, 1999) using primers against GFP (see below). For RT product measurements, DNA was extracted from $5 \times 10^5$ infected cells using the DNeasy Blood & Tissue kit (Qiagen) according to the manufacturer's protocol. DNA concentration was quantified using a Nanodrop for normalisation. RT products were quantified by TaqMan qPCR using TaqMan Gene Expression Master Mix (Thermo Fisher) and primers and probe specific to GFP. A dilution series of plasmid encoding GFP was measured in parallel to generate a standard curve to calculate the number of GFP copies.

*GFP* fwd: 5′- CAACAGCCACAACGTCTATATCAT -3′
*GFP* rev: 5′- ATGTTGTGGCGGATCTTGAAG -3′
*GFP* probe: 5′- FAM-CCGACAAGCAGAAGAACGGCATCAA-TAMRA -3′

### Infection assays

THP-1 cells were infected at a density of $2 \times 10^5$ cells/ml. For differentiation, THP-1 cells were treated with 50 ng/ml phorbol 12-myristate 13-acetate (PMA, PeproTech) for 48 h. Luciferase reporter assays were performed in 24-well plates and qPCR and ELISA in 12-well plates. Infection levels were assessed at 48 h post-infection through enumeration of GFP-positive cells by flow cytometry. Infections in THP-1 cells were performed in the presence of polybrene (8 µg/ml, Sigma). Input dose of virus was normalised either by RT activity (measured by SG-PERT) or genome copies (measured by qPCR) as indicated.

### Luciferase reporter assays

Gaussia/Lucia luciferase activity was measured by transferring 10 µl supernatant to a white 96-well assay plate, injecting 50 µl per well of coelenterazine substrate (Nanolight Technologies, 2 µg/ml) and analysing luminescence on a FLUOstar OPTIMA luminometer (Promega). Data were normalised to a mock-treated control to generate a fold induction.

### ISG qPCR

RNA was extracted from THP-1/primary MDM using a total RNA purification kit (Norgen) according to the manufacturer's protocol. Five hundred ng RNA was used to synthesise cDNA using Superscript III reverse transcriptase (Invitrogen), also according to the manufacturer's protocol. cDNA was diluted 1:5 in water and 2 µl was used as a template for real-time PCR using SYBR® Green PCR master mix (Applied Biosystems) and a Quant Studio 5 real-time PCR machine (Applied Biosystems). Expression of each gene was normalised to an internal control (*GAPDH*), and these values were then normalised to mock-treated control cells to yield a fold induction. The following primers were used:
*GAPDH:* Fwd 5′-GGGAAACTGTGGCGTGAT-3′, Rev 5′-GGAGGAGT GGGTGTCGCTGTT-3′
*CXCL-10:* Fwd 5′-TGGCATTCAAGGAGTACCTC-3′, Rev 5′-TTGTAGC AATGATCTCAACACG-3′
*IFIT-2:* Fwd 5′-CAGCTGAGAATTGCACTGCAA-3′, Rev 5′-CGTAGGC TGCTCTCCAAGGA-3′
*MxA:* Fwd 5′-ATCCTGGGATTTTGGGGCTT-3′, Rev 5′-CCGCTTGTC GCTGGTGTCG-3′

*CXCL-2:* Fwd 5′-GGGCAGAAAGCTTGTCTCAA-3′, Rev 5′-GCTTCCT CCTTCCTTCTGGT-3′

### ELISA

Cell supernatants were harvested for ELISA at 24 h post-infection/stimulation and stored at −80°C. CXCL-10 protein was measured using DuoSet ELISA reagents (R&D Biosystems) according to the manufacturer's instructions.

### Immunoblotting

For immunoblotting of viral particles, $2 \times 10^{11}$ genome copies of virus were boiled for 10 min in 6× Laemmli buffer (50 mM Tris–HCl (pH 6.8), 2% (w/v) SDS, 10% (v/v) glycerol, 0.1% (w/v) bromophenol blue, 100 mM β-mercaptoethanol) before separating on 4–12% Bis-Tris polyacrylamide gradient gel (Invitrogen). For immunoblot analysis of THP-1 cells, $3 \times 10^6$ cells were lysed in a cell lysis buffer containing 50 mM Tris pH 8, 150 mM NaCl, 1 mM EDTA, 10% (v/v) glycerol, 1% (v/v) Triton X-100, 0.05% (v/v) NP40 supplemented with protease inhibitors (Roche), clarified by centrifugation at $14,000 \times g$ for 10 min and boiled in 6× Laemmli buffer for 5 min. Proteins were separated by SDS-PAGE on 12% polyacrylamide gels. After PAGE, proteins were transferred to a Hybond ECL membrane (Amersham biosciences) using a semi-dry transfer system (Bio-Rad). Primary antibodies were from the following sources: mouse anti-β-actin (Abcam), rabbit-anti-SAMHD1 (ProteinTech) and mouse anti-HIV-1 capsid p24 (183-H12-5C, AIDS Reagents). Primary antibodies were detected with goat-anti-mouse/rabbit IRdye 800CW infrared dye secondary antibodies and membranes imaged using an Odyssey Infrared Imager (LI-COR Biosciences).

### Abrogation-of-restriction assay

FRhK4 cells were plated in 48-well plates at $5 \times 10^4$ cells/ml. The following day cells were co-transduced in the presence of polybrene (8 µg/ml, Sigma) with a fixed dose of HIV-1 GFP ($5 \times 10^7$ genome copies/ml) and increasing doses of HIV-LUC ΔCA-SP1 mutants or LPV-treated HIV-LUC viruses ($1.7 \times 10^6$–$3.8 \times 10^9$ genome copies/ml). Rescue of GFP infectivity was assessed 48 h later by flow cytometry using the FACS Calibur (BD) and analysing with FlowJo software.

### Statistical analyses

Statistical analyses were performed using an unpaired Student's *t*-test (with Welch's correction where variances were unequal) or a two-way ANOVA with multiple comparisons, as indicated. $*P < 0.05$, $**P < 0.01$, $***P < 0.001$.

## Data availability

This study includes no data deposited in external repositories.

**Expanded View** for this article is available online.

## Acknowledgements

We thank Veit Hornung for kindly providing THP-1-IFIT-1 cells. This work was funded through a Wellcome Trust Senior Biomedical Research Fellowship (GJT), the European Research Council under the European Union's Seventh Framework Programme (FP7/2007-2013)/ERC (grant HIVInnate 339223) and the National Institute for Health Research University College London Hospitals Biomedical Research Centre and a Wellcome Trust Collaborative award.

## Author contributions

RPS and GJT conceived the study. RPS, LH, TPP, ET, MS and LZ-A performed the experiments. RPS, LH, TPP and GJT analysed the data. RPS and GJT wrote the manuscript.

## Conflict of interest

The authors declare that they have no conflict of interest.

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
