## [Review Process File · The EMBO Journal]

Disrupting HIV-1 capsid formation causes cGAS sensing of viral DNA

Rebecca Sumner, Lauren Harrison, Emma Touizer, Thomas Peacock, Matthew Spencer, Lorena Zuliani-Alvarez, and Greg Towers

DOI: [10.15252/embj.2019103958](https://doi.org/10.15252/embj.2019103958)

Corresponding author(s): Rebecca Sumner (r.sumner@ucl.ac.uk)

Review Timeline:

Submission Date:	8th Nov 19
Editorial Decision:	15th Jan 20
Revision Received:	18th May 20
Editorial Decision:	2nd Jul 20
Revision Received:	23rd Jul 20
Accepted:	27th Jul 20

Editor: Karin Dumstrei

Transaction Report:

Dear Rebecca,

Thank you for sending your point-by-point response to the comments raised by the referees. I have now had a chance to take a careful look at it. I appreciate your response and would like to ask you address the referees' concerns as outlined in your response. Let me know if we need to discuss anything further.

Thank you for the opportunity to consider your work for publication. I look forward to your revision.

with best wishes

Karin

Karin Dumstrei, PhD
Senior Editor
The EMBO Journal

When assembling figures, please refer to our figure preparation guideline in order to ensure proper formatting and readability in print as well as on screen:
<http://bit.ly/EMBOPressFigurePreparationGuideline>

- a point-by-point response to the referees' comments, with a detailed description of the changes made (as a word file).
- a word file of the manuscript text.

- individual production quality figure files (one file per figure)
 - a complete author checklist, which you can download from our author guidelines (<https://www.embopress.org/page/journal/14602075/authorguide>).
 - Expanded View files (replacing Supplementary Information)
- Please see out instructions to authors
<https://www.embopress.org/page/journal/14602075/authorguide#expandedview>

The revision must be submitted online within 90 days; please click on the link below to submit the revision online before 14th Apr 2020.

Link Not Available

Referee #1:

General Summary

In this manuscript, Sumner RP et al., studied the role of viral capsid in determining innate immune sensing of HIV via the cGAS STING pathways. For that, they have employed genetic and pharmacological strategies to disrupt the integrity or maturation of viral capsid. They confirm that by encapsulation of viral cDNA prevents activation of the cGAS-STING innate immune pathway. Overall the experiments look well conducted. The main concern is the conceptual novelty. The role of viral capsid in innate immune evasion by HIV and more specifically recognition via the cGAS-STING signaling has been reported (e.g. Lahaye X et al., Immunity 2013 Dec 12;39(6):1132-42).

Main concerns/suggestions

- 1) A major assumption of the author's proposed model is that recognition of HIV cDNA by cGAS occurs in the cytosol. However recent studies show that cGAS is in fact abundant in the nucleus (e.g. Gentili, M et al Cell reports 2019, Jiang H et al EMBO J 2019) and that recognition of HIV cDNA by cGAS likely occurs in the nucleus (Lahaye X et al., Cell 2018). Given that the main conclusion of the manuscript in the current form is not really novel, to move the field forward, I suggest the authors do experiments to address the issue whether the observed cGAS-mediated innate immune recognition occur in the cytosol or the nucleus by testing the effect of nuclear import inhibitors on cGAS activation by HIV.
- 2) Clarity: The manuscript is not clearly written. The main text is cluttered with unnecessary information making it hard to read and grasp the key points. To improve clarity, much of the information in the Results and Figure legends sections can easily be moved to the Materials and Method section.

Referee #2:

In this study, the authors have investigated the impact of interfering with HIV-1 capsid maturation on innate sensing in myeloid cells. They show that HIV-1 stocks produced in the presence of sub-optimal doses of protease (PR) inhibitors trigger an IFN response in SAMHD1-depleted THP-1 cells, and CXCL10 expression in monocyte-derived macrophages (MDMs). A similar observation is made on these THP-1 cells when viral stocks are made of hybrid viruses with partially deficient mutant capsid. They show that the innate immune activation requires the viral DNA and the cGAS-STING pathway. Finally, the capsid-destabilizing drug PF-74 is shown to promote innate sensing in SAMHD1-depleted THP-1 cells following HIV infection.

The quality of the data is very high, the experiments are very well controlled and carefully executed. The authors have made an excellent job at thoroughly quantifying and normalizing the HIV preparations with altered Gag processing, enabling robust comparison with the WT controls. The paper is well written and the figures are clear. These are important strengths of the paper.

A current limitation of the study is that almost all the data was generated in PMA-treated THP-1 cells (a cancer cell line, not a natural target of HIV-1) that are additionally knocked-down for SAMHD1 (a situation that normally only applies to Vpx+ viruses such as SIVmac or HIV-2), instead of primary CD4+ cells. This is important because interactions between viruses and innate sensors are the results of co-evolution, by necessity within the relevant target cells that viruses infect. Cancer cells in particular have evolved to deregulate innate sensors (PMID 30846571). This implies that virus-sensor interactions observed in cancer cells may not necessarily apply to primary target cells. The results with PF-74 are problematic in this regard because they previously convincingly showed that this treatment does not induce IFN in primary macrophage (Rasaiyaah et al. Nature 2013).

The significance of the findings is also not evident. Conceptually, the new study provides an original validation of the working model, but it is already genetically established that HIV-1 WT capsid cloaks the viral DNA, thereby limiting sensing by cGAS (Rasaiyaah et al. 2013 and Lahaye et al. 2013). Clinically, it is difficult to imagine how to apply sub-optimal doses of LPV in monotherapy. This would presumably work at a narrow dose range of LPV, sufficient to limit Gag processing, but not high enough to block viral DNA synthesis. It is not evident whether this could be reached in certain anatomical regions or at a certain time after treatment (i.e. PK/PD). Furthermore, vDNA is required to activate cGAS, so patients should be on monotherapy with low dose LPV - this does not sound feasible from a clinical perspective. Presumably, the better clinical strategy remains to use higher doses of PR inhibitor, in combination therapy, to block replication in patients.

Overall, the study revisits the question DNA cloaking by capsid through an original and interesting angle (capsid assembly and stability). The results constitute a solid proof-of-concept in artificial settings that sub-optimal interference with HIV-1 capsid formation can favor innate sensing of the virus. The study is interesting but the significance is currently limited to a modified cancer cell line.

1. It would strengthen the study if they could directly visualize disassembly of the viral capsid in the cytoplasm after RT (WT vs LPV or CA-SP1 or PF74) and demonstrate that this promotes direct association of the viral DNA with cGAS (IP-qPCR).
2. The significance of the findings is essentially limited to THP-1 cells knocked-down for SAMHD1.

THP-1 is a cancer cell line, which unfortunately does not represent natural target cells of the virus. Stable SAMHD1 knock-down is an additional concern, since it was reported that SAMHD1 may play a role in the differentiation of THP-1 into macrophages by PMA, as used here (PMID 26606981). Only Fig 1K/L was generated in monocyte-derived macrophages. It is difficult to accept their broad claim that "disrupting capsid formation causes cGAS sensing of viral DNA" (title), based on a single piece of data in primary target cells of the virus. Can they show that CA-SP1 mixed virus and PF74 also induces an IFN and ISG response in MDMs (and not only CXCL10)? In primary CD4+ T cells?

3. Fig 1K: l. 164-167 In MDM, which the important cell type here, only CXCL10 was measured. CXCL10 can be induced by many other pathways (such as NFkB) in the absence of IFN. It is not enough to claim that an IFN response is induced (l.167). They need to provide direct evidence that IFN is expressed and bioactive (qPCR, ELISA), and that several ISG are induced (qPCR, western blots), in MDM.

4. According to the legend, it appears that in all figures, statistical tests are applied to technical replicates within one experiment. This does not inform on biological reproducibility of the findings. They should instead systematically aggregate the data from the independent biological experiments, and perform the statistical tests on the biological replicates. This would also better address inter-experiment variation discussed in l.216-218, for example.

5. In their previous Nature 2013 paper (Rasaiyaah et al.), they showed that 10 μ M PF-74 does not lead to an IFN response after HIV-1 infection of MDM. Now, they obtain the opposite result in THP-1 shSAMHD1 cells. This strengthens my concern that these cells do not constitute a faithful model of primary target cells.

Other comments:

6. In the experiments with LPV, suboptimal Gag processing means sub-optimal Gag-Pol processing, thus sub-optimal amounts of free RT. Since they normalize virus preps by SG-PERT assay and compare viruses at similar doses of RT (0.1U or 0.5U), there is a possibility that they have over-estimated the physical quantity of viral particles in the LPV treated samples. What are the p24 concentrations or genome copy numbers in these conditions? Can they discuss this point?

7. In the introduction l.47-47, an important intracellular PRR for HIV-1 to mention is NONO (PMID 30270045). Subsequently, l.61-64, it should also be mentioned that MDMs sense HIV-1 through the NONO-cGAS pathway if the SAMHD1 restriction is abrogated (PMID 30270045). This is relevant here to justify the depletion of SAMHD1 in all the THP-1 experiments.

8. Abstract l.21-22: There is a confusion here. HIV-1 is a strong inducer of IFN in infected patients, as demonstrated by high levels of IFN in the circulation (PMID 25505958). pDCs also induce massive amounts of IFN in response to HIV-1. However, the virus does not induce high levels of IFN when it replicates in macrophages or lymphocytes. Either specify the relevant target cell types, or restrict to 'intracellular' or 'replication-associated' induction.

9. Abstract l.22-23: The concept that the HIV-1 capsid cloaks the viral DNA from cGAS has already been proposed in Rasaiyaah et al. 2013 and in Lahaye et al. 2013. It is odd to present this as a new hypothesis. Please reformulate to clarify the new and more restricted hypothesis here related to capsid assembly and processing.

10. The word 'capsid' is used interchangeably when referring to the viral capsid protein (e.g. l.62) or the viral capsid itself (i.e. the assembled structure made of individual capsid proteins, e.g. l.72). This is confusing, and important in the context of this particular study, please make it more explicit throughout.

11. Fig 5: TRIM5a blocks HIV before reverse transcription, but RT is a requisite for sensing in THP-1 cells here. These results appear confirmatory for TRIM5a antiviral activity (l.288) and only distantly related to the question of DNA cloaking.

12. Suppl. Figures, and panels, should be organized and numbered according to order to appearance in the text.

13. Fig 11.152-153: the data is on a log-scale and I think the vDNA decreases with increasing dose of LPV by at least 3x. Instead of 'was not changed' perhaps write instead that the vDNA levels do not 'increase' with LPV?

14. l.130-131 Cingoz & Goff did not use primary macrophages, but normal human dermal fibroblasts, that are not CD4+ and not targets of HIV-1, please correct. Other studies demonstrated that HIV-1 can induce an ISG response in THP-1 cells (PMID 23929945) and more importantly in primary macrophages (PMID 28490595), they should be cited and considered for discussion.

15. L.342-351 It is unsettling that they make a point about the suppression of SAMHD1 in primary cells used in other studies, since they have used SAMHD1 knock-down cells for 99% of the present study, apparently. In primary MDM, they have only measured CXCL10 in response to LPV-treated viral stocks. I do not think that they can broadly claim from this single data that disrupting HIV-1 capsid confirmation causes cGAS sensing (title) in primary macrophages.

In PMID 28490595 purified HIV-1 viral stock induced an ISG response in macrophages, without manipulation of TREX1 or complementation with Vpx. This should be mentioned, and the possible differences discussed.

Related to this, "We and others have found that at very high dose, HIV-1 activates innate immune pathways and this is influenced particularly by whether the viral supernatant is purified" is a rather vague statement. Who are "others"? How is a "very high dose" defined? One could argue that they used a high dose in this study as well: in Fig 2B, they have 50-60% infected cells, which is MOI>1. In Fig 6, they see an ISG response mainly at 3 U/ml of virus, a dose at which the %GFP+ cells already decays (Fig 6B), again indicating MOI>1 on target cells.

Invoking purification as a confounding factor seems also highly unlikely, since in most of the cited papers l.342-348, reverse transcription was required for sensing, thus excluding sensing of contaminating non-viral material.

Arguably, the authors raise an important and valid question, but it would strengthen the discussion if they could make a more factual analysis in this section, instead of what seems like opinions.

16. In Fig 6, it is unclear if SAMHD1 knocked-down cells were used?

17. The title is too broad. Most of the data has been performed in THP-1 cells, and one experiment in MDM. This should be made clear in the title.

Referee #3:

In this manuscript from Sumner et al provide evidence that the structure of the HIV-1 capsid protects nascent viral reverse transcripts from cytoplasmic DNA sensing. Specifically, they show that viruses produced in the presence of low doses of protease inhibitors, capsid-destablizing drugs, or incorporation of gag proteins with mutations in the CA-SP1 PR cleavage site results in viruses that trigger cGAS/STING dependent interferon responses upon sensing of RT products in myeloid cell lines and primary macrophages.

This is an interesting, well-written and important manuscript. The authors have very carefully controlled their experiments to rule out contaminating DNA and other IFN-inducing impurities in their viral stocks, something that is sadly not the case for many high-profile papers published in this field. Thus, the authors make a compelling case that manipulating Gag processing at levels that have small effects on one-round viral infectivity, leads to exposure of reverse transcripts to cGAS/STING.

I have a few suggestions to enhance the general interest of the manuscript and the interpretation of key results that the authors could consider:

- The implication from much of the data is that low doses of PIs, at around the IC50, can cause the trigger IFN. It would be nice to try and put some of these observations into a more physiologically relevant context - i.e. to provide evidence that the efficacy of protease inhibitors is enhanced by the triggering of antiviral responses, which is something the authors skirt around without really addressing. Somethings that authors could consider are the following.
 - o The use predominantly of one-round virus infection throughout the MS, as the authors rightly point out, would not be expected to reveal an antiviral IFN phenotype. However, a spreading infection in human macrophages may do so. The prediction would be that inhibition of replication in macrophages in the presence of low-dose, but not high-dose, PI concentrations would be at least partially interferon dependent.
 - o PI resistance in PR comes at a fitness cost that leads to second site mutations in the Gag cleavage sites. Do PI resistance mutations in PR trigger more IFN in macrophages/THP-1 and if so to the second-site mutations in Gag rescue it?
- The authors suggest that CA structure is compromised, but in reality the 30nM treatment of LPV is sufficient to trigger the majority of the IFN, and this is still fully capable of saturating T5a in FRhK. Thus, these viruses are sufficiently wt in shape to bind TRIM5, implying the compromisation is more subtle. Capsid stability seems an obvious thing to look at - is there any further information the authors can add on this score.
- WBs for STING, cGAS and MAVS in the KO cells (and CRISPR lesions) should be included.
- The MAVS observation with DRV is weird. Is it incoming viral gRNA or virion associated RNAs?

Referee #1

General Summary

In this manuscript, Sumner RP et al., studied the role of viral capsid in determining innate immune sensing of HIV via the cGAS STING pathways. For that, they have employed genetic and pharmacological strategies to disrupt the integrity or maturation of viral capsid. They confirm that by encapsulation of viral cDNA prevents activation of the cGAS-STING innate immune pathway. Overall the experiments look well conducted. The main concern is the conceptual novelty. The role of viral capsid in innate immune evasion by HIV and more specifically recognition via the cGAS-STING signaling has been reported (e.g. Lahaye X et al., Immunity 2013 Dec 12;39(6):1132-42).

Main concerns/suggestions

1) A major assumption of the author's proposed model is that recognition of HIV cDNA by cGAS occurs in the cytosol. However recent studies show that cGAS is in fact abundant in the nucleus (e.g. Gentili, M et al Cell reports 2019, Jiang H et al EMBO J 2019) and that recognition of HIV cDNA by cGAS likely occurs in the nucleus (Lahaye X et al., Cell 2018). Given that the main conclusion of the manuscript in the current form is not really novel, to move the field forward, I suggest the authors do experiments to address the issue whether the observed cGAS-mediated innate immune recognition occur in the cytosol or the nucleus by testing the effect of nuclear import inhibitors on cGAS activation by HIV.

The question of exactly where sensing by cGAS occurs is certainly important but it is quite intractable with current methods. A recent paper from the Manel lab (PMID: 30270045) suggested that HIV-2 cDNA could be recognised by cGAS in the nucleus, dependent on an interaction between the nuclear protein NONO and the HIV-2 capsid. The HIV-1 capsid however only bound NONO very weakly and it was unclear whether nuclear sensing of HIV-1 DNA occurred. How one might distinguish between cytoplasmic and nuclear sensing is unclear. As far as we are aware there are no HIV-1-specific nuclear import inhibitors, as suggested by the reviewer, and any non-specific nuclear import inhibitors would be expected to impact nuclear import of the IFN activated transcription factors thereby preventing measuring of sensing. We propose that working out whether cGAS sensing occurs in the nucleus or cytoplasm is beyond the scope of this study. We have adjusted the introduction to include the following text:

'The nuclear protein NONO has also been implicated in the detection of HIV-2 cDNA (Lahaye et al, 2018)' (line 52) and ... 'and HIV-1/2 can be sensed by a process requiring NONO if restriction by SAMHD1 is overcome (Lahaye et al., 2018)' (line 74).

We have also added to the discussion:

'Recent work has suggested that detection of HIV-2 viral cDNA by cGAS can occur in the nucleus of infected dendritic cells and macrophages, with a role for interaction of the nuclear protein NONO and the viral capsid (Lahaye et al., 2018). Concordantly, recent studies suggest that HIV-1 uncoats in the nucleus, rather than as previously thought, in the cytoplasm (Bejarano et al, 2019; Burdick et al., 2017; Burdick et al, 2020; Francis & Melikyan, 2018). Here we propose that PI inhibited HIV-1 is detected in the cytoplasm because defective viral capsids are expected to fail in the cytoplasm, before they reach the nucleus, but this has not formally been proven. Further work will address whether nuclear HIV DNA can be detected by cGAS and how nuclear cGAS avoids detecting cellular DNA.' (lines 545-554).

We have also clarified the novelty of this work in lines 492-495 of the discussion: 'The work presented herein suggests that, regardless of cofactor and restriction factor interactions, the HIV-1 viral capsid also plays a protective role and physically protects viral DNA from the sensor cGAS through a process of encapsidated DNA synthesis, which is disturbed by disturbing Gag cleavage.'

2) Clarity: The manuscript is not clearly written. The main text is cluttered with unnecessary information making it hard to read and grasp the key points. To improve clarity, much of the information in the Results and Figure legends sections can easily be moved to the Materials and Method section.

We have examined the text again and made a series of minor changes that we hope will improve clarity. Regarding information that should be in methods, we were keen to point out how we have standardised viral dose in the main text, for example, because it is so important for interpretation of these experiments, as reiterated by referee 2. We have however shortened the figure legends to remove information that can be found in the methods.

Referee #2

In this study, the authors have investigated the impact of interfering with HIV-1 capsid maturation on innate sensing in myeloid cells. They show that HIV-1 stocks produced in the presence of sub-optimal doses of protease (PR) inhibitors trigger an IFN response in SAMHD1-depleted THP-1 cells, and CXCL10 expression in monocyte-derived macrophages (MDMs). A similar observation is made on these THP-1 cells when viral stocks are made of hybrid viruses with partially deficient mutant capsid. They show that the innate immune activation requires the viral DNA and the cGAS-STING pathway. Finally, the capsid-destabilizing drug PF-74 is shown to promote innate sensing in SAMHD1-depleted THP-1 cells following HIV infection.

The quality of the data is very high, the experiments are very well controlled and carefully executed. The authors have made an excellent job at thoroughly quantifying and normalizing the HIV preparations with altered Gag processing, enabling robust comparison with the WT controls. The paper is well written and the figures are clear. These are important strengths of the paper.

A current limitation of the study is that almost all the data was generated in PMA-treated THP-1 cells (a cancer cell line, not a natural target of HIV-1) that are additionally knocked-down for SAMHD1 (a situation that normally only applies to Vpx+ viruses such as SIVmac or HIV-2), instead of primary CD4+ cells. This is important because interactions between viruses and innate sensors are the results of co-evolution, by necessity within the relevant target cells that viruses infect. Cancer cells in particular have evolved to deregulate innate sensors (PMID 30846571). This implies that virus-sensor interactions observed in cancer cells may not necessarily apply to primary target cells. The results with PF-74 are problematic in this regard because they previously convincingly showed that this treatment does not induce IFN in primary macrophage (Rasaiyaah et al. Nature 2013).

The significance of the findings is also not evident. Conceptually, the new study provides an original validation of the working model, but it is already genetically established that HIV-1 WT capsid cloaks the viral DNA, thereby limiting sensing by cGAS (Rasaiyaah et al. 2013 and Lahaye et al. 2013). Clinically, it is difficult to imagine how to apply sub-optimal doses of LPV in monotherapy. This would presumably work at a narrow dose range of LPV, sufficient to limit Gag processing, but not high enough to block viral DNA synthesis. It is not evident whether this could be reached in certain anatomical regions or at a certain time after treatment (i.e. PK/PD). Furthermore, vDNA is required to activate cGAS, so patients should be on monotherapy with low dose LPV - this does not sound feasible from a clinical perspective. Presumably, the better clinical strategy remains to use higher doses of PR inhibitor, in combination therapy, to block replication in patients.

Overall, the study revisits the question DNA cloaking by capsid through an original and interesting angle (capsid assembly and stability). The results constitute a solid proof-of-concept in artificial settings that sub-optimal interference with HIV-1 capsid formation can favor innate sensing of the virus. The study is interesting but the significance is currently limited to a modified cancer cell line.

1. It would strengthen the study if they could directly visualize disassembly of the viral capsid in the cytoplasm after RT (WT vs LPV or CA-SP1 or PF74) and demonstrate that this promotes direct association of the viral DNA with cGAS (IP-qPCR).

This is an interesting point, and in response, we began to optimise a cGAS IP followed by qPCR for the viral DNA as was performed by the Manel lab in their recent NONO publication (PMID: 30270045). Unfortunately our lab closed in March preventing us from completing this experiment. Given that this experiment would not significantly change the conclusions of the manuscript we hope that it is not required. We have provided more evidence for our findings and model as detailed below.

2. The significance of the findings is essentially limited to THP-1 cells knocked-down for SAMHD1. THP-1 is a cancer cell line, which unfortunately does not represent natural target cells of the virus. Stable SAMHD1 knock-down is an additional concern, since it was reported that SAMHD1 may play a role in the differentiation of THP-1 into macrophages by PMA, as used here (PMID 26606981). Only Fig 1K/L was generated in monocyte-derived macrophages. It is difficult to accept their broad claim that "disrupting capsid formation causes cGAS sensing of viral DNA" (title), based on a single piece of data in primary target cells of the virus. Can they show that CA-SP1 mixed virus and PF74 also induces an IFN and ISG response in MDMs (and not only CXCL10)? In primary CD4+ T cells?

We agree that the manuscript is strengthened by including additional experiments in primary macrophages. We have now included data demonstrating that HIV-1 GFP Δ CA-SP1 (75 % mutant) also induces an enhanced ISG response in MDM, including qPCR for CXCL-10, IFIT-2 and MxA (Fig. 2I), as well as CXCL-10 protein secretion by ELISA (Fig. 2K). This ISG induction was dependent on IFN secretion, evidenced by this being significantly reduced by the JAK inhibitor ruxolitinib (Fig. 2J). Associated text changes are as follows (line 280):

'To corroborate these findings in primary cells, we infected MDM with HIV-1 GFP Δ CA-SP1 (75 % mutant) and found enhanced CXCL-10, IFIT-2 and MxA expression compared to WT HIV-1 GFP (Fig. 2I, EV2D). Furthermore HIV-1 GFP Δ CA-SP1 induced an IFN response in these cells, as treatment with ruxolitinib significantly reduced IFIT-2 expression (Fig. 2J) and CXCL-10 secretion (Fig. 2K) induced by HIV-1 GFP Δ CA-SP1. Interestingly in primary MDM treatment of cells with ruxolitinib did enhance infection levels of HIV-1 GFP Δ CA-SP1, but not WT HIV-1 GFP. This is consistent with the notion that HIV-1 GFP Δ CA-SP1 induces a IFN-dependent antiviral response in these cells that is, in this case, fast enough to inhibit single round infection (Fig. EV2E, F).'

Measuring innate immune responses in primary CD4+ T cells is difficult to interpret as unfortunately HIV does not replicate in T cells unless they are activated, typically by TCR cross linking. This causes massive TCR mediated signalling which likely overwhelms innate immune activation by virus, where it exists. For this reason we have not examined infection in T cells. Ongoing work seeks to develop infection assays without T cell activation and these points can be addressed when we have this working. We hope the reviewers agree that, for these reasons T cell work is beyond the scope of the study. We have added the following text to the discussion to make this point (line 600).

"In this study we have focused on infection of macrophages and macrophage-like THP-1 cells. Unfortunately HIV does not replicate in primary human T cells, unless they are activated, typically by cross-linking and activating the T cell receptor (TCR). This causes massive TCR mediated signalling which is likely to overwhelm T cell innate immune signalling driven by infection. The study of signalling induced by HIV therefore awaits the development of tractable infection T cell infection models that do not require TCR activation.

3. Fig 1K: l. 164-167 In MDM, which the important cell type here, only CXCL10 was measured. CXCL10 can be induced by many other pathways (such as NFkB) in the absence of IFN. It is not enough to claim that an IFN response is induced (l.167). They need to provide direct evidence that IFN is expressed and bioactive (qPCR, ELISA), and that several ISG are induced (qPCR, western blots), in MDM.

As described above in response to point 2 we have now included data from primary MDM showing activation of several ISGs (Fig. 2I). We have also included data to demonstrate these viruses induce a bioactive IFN response through the use of the JAK inhibitor ruxolitinib that blocks IFN signalling. Data in Fig. 2J and K demonstrate that the ISG response in primary MDM to HIV-1 GFP Δ CA-SP1 (75 % mutant) is IFN dependent and new data provided in Fig. EV1K also show that DRV-treated

HIV-1 GFP induces CXCL-10 secretion in primary MDM in an IFN-dependent manner. We have attempted to measure soluble IFN in supernatants from THP-1 cells and primary MDM by ELISA and bioassay using 293 cells that have a stable luciferase reporter driven by an ISRE, however we have consistently failed to detect anything. Either the amount of IFN secretion is below the limit of detection in these experiments or the time points we have tested were not appropriate. However, further evidence that bioactive IFN is secreted from these cells is also demonstrated in Fig. 2E & F where HIV-1 GFP Δ CA-SP1 (75 % mutant) infection in primary MDM is rescued by IFN blockade.

Associated text changes to describe new data in Fig EV1 are as follows (line 208):

'Similarly, DRV-treated HIV-1 GFP induced more CXCL-10 secretion in primary MDM than untreated HIV-1 GFP (0 nM DRV) and this was dependent on type I IFN production, as evidenced by the lack of CXCL-10 production in the presence of ruxolitinib (Fig. EV1K). Infection levels were not changed by ruxolitinib treatment (Fig EV1L). Together, these data suggest that infection by PI-treated HIV-1 induces an IFN-dependent innate immune response in PMA-treated THP-1 cells and primary human MDM that is not observed after infection with untreated virus.'

Some additional discussion has also been added (line 569):

'Conversely infection by the 75% Δ CA-SP1 HIV-1 mutant in primary MDM was rescued by ruxolitinib (Fig. EV2E, F). Thus this virus, made by mixing WT and mutant Gag constructs, causes IFN production that subsequently inhibits viral infection. We expect that primary MDM may exhibit a faster or more potent IFN response allowing IFN mediated suppression of infection even during single round infection of these cells. This result is reminiscent of rescue of infectivity, with IFN receptor blockade, of HIV-1 bearing CA mutants P90A and N74D in MDM (Rasaiyaah et al., 2013). We hypothesise that whether IFN inhibition rescues infection or not depends on the degree to which the suppression of replication depends on IFN secretion. For example, IFN inhibition does not rescue replication in PI-treated MDM because protease inhibition is sufficient for the observed viral inhibition. Conversely, in the case of Gag cleavage mutants (Fig. EV2E, F), single round infection is in part inhibited by IFN and thus, JAK/STAT inhibition rescues some degree of infectivity. In vivo, whether the PI inhibited virus is directly suppressed by IFN or not, we would expect IFN to contribute to innate and adaptive immune responses and globally contribute to viral inhibition.'

4. According to the legend, it appears that in all figures, statistical tests are applied to technical replicates within one experiment. This does not inform on biological reproducibility of the findings. They should instead systematically aggregate the data from the independent biological experiments, and perform the statistical tests on the biological replicates. This would also better address inter-experiment variation discussed in 1.216-218, for example.

In our experience the magnitude of innate activation between experiments can vary greatly making collation of data from multiple experiments noisy and difficult to analyse. All of our experiments have been performed three or more times, and at least twice with consistent biological triplicates as a minimum, from which statistical analyses have been performed. We have made this clearer in the figure legends, for example 'Data are mean \pm SD, n=3, representative of 3 repeats.'

5. In their previous Nature 2013 paper (Rasaiyaah et al.), they showed that 10 μ M PF-74 does not lead to an IFN response after HIV-1 infection of MDM. Now, they obtain the opposite result in THP-1 shSAMHD1 cells. This strengthens my concern that these cells do not constitute a faithful model of primary target cells.

The discrepancy in data obtained in the 2013 Nature paper and the current manuscript are likely explained by differences in experimental conditions rather than the cell types per se. The dose of PF-74 used in the primary MDM in our previous work (10 μ M) was sufficient to inhibit viral infection in these cells, as well as cDNA synthesis, thus no PAMP was present to stimulate an IFN response. In the experiments presented in figure 6 of this manuscript in THP-1 cells, 10 μ M PF-74 was not

sufficient to inhibit viral infection at the higher doses of virus (see Fig 6B), thus viral PAMP is made and cGAS is activated. We have added the following text to make this point, line 436:

'These results in THP-1 differ from our previous observation in MDM (Rasaiyaah et al., 2013), in which PF-74 treatment did not induce sensing of HIV-1. We assume that the 10 μ M PF-74 used in MDM inhibited viral DNA (PAMP) synthesis, preventing cGAS activation.'

Other comments:

6. In the experiments with LPV, suboptimal Gag processing means sub-optimal Gag-Pol processing, thus sub-optimal amounts of free RT. Since they normalize virus preps by SG-PERT assay and compare viruses at similar doses of RT (0.1U or 0.5U), there is a possibility that they have over-estimated the physical quantity of viral particles in the LPV treated samples. What are the p24 concentrations or genome copy numbers in these conditions? Can they discuss this point?

We appreciate the point that consideration of dose is essential in this study. The maximal difference in RT activity between viruses was 5 fold for 100nM LPV (line 164 of the text) and only 2-fold for 30nM LPV. Genome copy measurements (i.e. particle numbers) differed by up to 2 fold for the viruses with higher doses of LPV. The difference in triggering observed for LPV-treated versus WT virus was an order of magnitude greater than this and signalling was found to be dependent on reverse transcription and the DNA sensing machinery, consistent with sensing of DNA positive particles. Importantly RT products in infected cells were no higher for the LPV-treated viruses indicating that the same amount of PAMP is detected only when PI treated (Fig. 11). We have included the following (line 162) for clarification:

'Virus dose in these experiments was normalised according to RT activity, as measured by SG-PERT (see Methods), which differed no more than 5-fold in the LPV-treated versus untreated virus. Determination of genome by qRT-PCR gave similar dose values.'

And also the following to the legend of figure 1:

'For experiments in which the virus dose used was normalised by RT activity, the number of genome copies was also measured by qPCR of virus. This gave dose equivalents of within 2-3 fold of RT equivalents.'

Of note, experiments with DRV-treated virus (Fig EV1), as well as experiments in later figures, were performed by normalising for genome copies to avoid any issues with differences in RT activity. The Δ CA-SP1 mutation does not affect RT activity and this has been clarified in line 252 as follows 'Virus dose in these experiments was normalised according to RT activity, which differed no more than 5-fold between viruses. Importantly, differences in RT activity, measured by SG-PERT, were mirrored by measurements of genome copy, measured by qPCR. This is consistent with variation in viral production rather than inhibition of RT activity by the Δ CA-SP1 mutation.' Lines 524-531 of the discussion also address issues of normalising viral doses.

7. In the introduction 1.47-47, an important intracellular PRR for HIV-1 to mention is NONO (PMID 30270045). Subsequently, 1.61-64, it should also be mentioned that MDMs sense HIV-1 through the NONO-cGAS pathway if the SAMHD1 restriction is abrogated (PMID 30270045). This is relevant here to justify the depletion of SAMHD1 in all the THP-1 experiments.

We have added references to NONO in the introduction as described above in response to reviewer 1 (lines 52 and 74).

8. Abstract 1.21-22: There is a confusion here. HIV-1 is a strong inducer of IFN in infected patients, as demonstrated by high levels of IFN in the circulation (PMID 25505958). pDCs also induce massive amounts of IFN in response to HIV-1. However, the virus does not induce high levels of IFN when it replicates in macrophages or lymphocytes. Either specify the relevant target cell types, or restrict to 'intracellular' or 'replication-associated' induction.

We have re-worded the abstract to:

'As HIV-1 replication is not a strong inducer of IFN we hypothesised that an intact capsid physically cloaks viral DNA from cGAS.' (line 22).

9. Abstract 1.22-23: The concept that the HIV-1 capsid cloaks the viral DNA from cGAS has already been proposed in Rasaiyaah et al. 2013 and in Lahaye et al. 2013. It is odd to present this as a new hypothesis. Please reformulate to clarify the new and more restricted hypothesis here related to capsid assembly and processing.

We have re-worded the abstract to: 'we hypothesised that an intact capsid physically cloaks viral DNA from cGAS.' (line 22).

10. The word 'capsid' is used interchangeably when referring to the viral capsid protein (e.g. 1.62) or the viral capsid itself (i.e. the assembled structure made of individual capsid proteins, e.g. 1.72). This is confusing, and important in the context of this particular study, please make it more explicit throughout.

This is a recurring problem in capsidology and we apologise for the lack of clarity. We have amended throughout to ensure it is clear whether we are referring to capsid protein or the whole viral capsid.

11. Fig 5: TRIM5a blocks HIV before reverse transcription, but RT is a requisite for sensing in THP-1 cells here. These results appear confirmatory for TRIM5a antiviral activity (1.288) and only distantly related to the question of DNA cloaking.

The data presented in Fig 5 in FRhK cells address the ability of these particles to form the authentic hexameric lattice required for recruitment of TRIM5a (line 379), thus informing on particle integrity. They do not address any TRIM5 signalling or innate sensing.

12. Suppl. Figures, and panels, should be organized and numbered according to order to appearance in the text.

We have ensured that all panels are in order of appearance in the text.

13. Fig 11.1.152-153: the data is on a log-scale and I think the vDNA decreases with increasing dose of LPV by at least 3x. Instead of 'was not changed' perhaps write instead that the vDNA levels do not 'increase' with LPV?

We have re-worded the text to say:

'Importantly, measurement of viral DNA production in infected PMA-treated THP-1 shSAMHDI cells, demonstrated that LPV did not increase DNA levels, ruling out increased DNA levels as an explanation for increased sensing (Fig. 11).' (line 193).

14. 1.130-131 Cingoz & Goff did not use primary macrophages, but normal human dermal fibroblasts, that are not CD4+ and not targets of HIV-1, please correct. Other studies demonstrated that HIV-1 can induce an ISG response in THP-1 cells (PMID 23929945) and more importantly in primary macrophages (PMID 28490595), they should be cited and considered for discussion.

Line 130, now line 159, refers to the THP-1 data in the Cingoz & Goff paper rather than primary cell data. We have however corrected lines 68-71 of the introduction to address the discrepancy pointed out by the reviewer: 'Work from our lab, and others, has demonstrated that primary monocyte-derived macrophages (MDMs) (Rasaiyaah et al., 2013; Tsang et al, 2009) and THP-1 cells (Cingoz & Goff, 2019) can be infected by wild-type (WT) HIV-1 without significant innate immune induction.'

The data from THP-1 cells in PMID 23929945 was obtained using Vpx co-transduction. Vpx is known to target cellular proteins other than SAMHDI and their potential roles in modulating innate immune

responses is currently unclear. In our experiments we have opted to deplete SAMHD1 rather than use Vpx-containing VLPs to enhance infection of PMA-differentiated THP-1 cells to avoid these issues. We have discussed this further in the discussion: 'For example, some studies have co-infected the cells with SIV VLPs bearing Vpx (Gao et al., 2013; Johnson et al, 2018; Manel et al, 2010; Yoh et al., 2015), which is known to degrade cellular proteins including SAMHD1, and the epigenetic regulator complex HUSH, and likely manipulates innate responses in complex ways.' (line 500). Addition of reference PMID 28490595 is addressed in response to point 15 below.

15. L.342-351 It is unsettling that they make a point about the suppression of SAMHD1 in primary cells used in other studies, since they have used SAMHD1 knock-down cells for 99% of the present study, apparently. In primary MDM, they have only measured CXCL10 in response to LPV-treated viral stocks. I do not think that they can broadly claim from this single data that disrupting HIV-1 capsid confirmation causes cGAS sensing (title) in primary macrophages.

As described above in response to point 14 other studies suppress SAMHD1 activity through co-transduction with Vpx-containing VLPs and Vpx is known to target cellular proteins other than SAMHD1, for example HUSH. This is why we chose to deplete SAMHD1 in the THP-1 cells rather than use Vpx. This issue has been clarified in the discussion (line 500). We have now included significantly more data in primary MDM showing that DRV-treated (Fig EV1K) and Δ CA-SP1 HIV-1 GFP (Fig. 2I, J, K) induce enhanced innate responses including measurements other than CXCL-10. Importantly, we did not manipulate SAMHD1 in the primary cell experiments which form a significant part of the work.

In PMID 28490595 purified HIV-1 viral stock induced an ISG response in macrophages, without manipulation of TREX1 or complementation with Vpx. This should be mentioned, and the possible differences discussed.

This is an interesting paper but quite different to ours. It demonstrates activation of IFN/ISGs by purified HIV-1, but in this study activation does not require viral genome or DNA synthesis. Thus this is not nucleic acid sensing and is instead sensing of viral entry (Env is required). We assume this different macrophage phenotype (capacity to detect entry) might be due to the different way the macrophages were produced, in this case by positive selection using beads, which may activate them. We have added this reference and discussed how different macrophage prep techniques may produce cells with different phenotypes in the amended manuscript (line 515): 'The method by which primary cells are purified and differentiated may also affect their activation status and hence sensing capacity. For example Decalf et al. describe activation of an IFN response in primary macrophages that was not dependent on reverse transcription or genomic RNA, attributing this response instead to detection of viral entry, dependent on the presence of a viral envelope (Decalf et al, 2017). In this study MDM were prepared by positive selection using CD14⁺ beads, which may activate the cells in ways that other purification methods do not.'

Related to this, "We and others have found that at very high dose, HIV-1 activates innate immune pathways and this is influenced particularly by whether the viral supernatant is purified" is a rather vague statement. Who are "others"? How is a "very high dose" defined? One could argue that they used a high dose in this study as well: in Fig 2B, they have 50-60% infected cells, which is MOI>1. In Fig 6, they see an ISG response mainly at 3 U/ml of virus, a dose at which the %GFP+ cells already decays (Fig 6B), again indicating MOI>1 on target cells.

Invoking purification as a confounding factor seems also highly unlikely, since in most of the cited papers 1.342-348, reverse transcription was required for sensing, thus excluding sensing of contaminating non-viral material.

Arguably, the authors raise an important and valid question, but it would strengthen the discussion if they could make a more factual analysis in this section, instead of what seems like opinions.

We have revisited this section of the discussion to include the publications suggested above by the reviewer and have provided a more thorough analysis of the publications referred to (paragraph

starting line 497): *'We propose that HIV-1 has evolved to cloak viral DNA synthesis within an intact capsid (Jacques et al., 2016; Rasaiyaah et al., 2013). However, several studies have reported innate immune activation by WT HIV-1 in macrophages or dendritic cells. There are various reasons why study results may differ. For example, some studies have co-infected the cells with SIV VLPs bearing Vpx (Gao et al., 2013; Johnson et al., 2018; Manel et al., 2010; Yoh et al., 2015), which is known to degrade cellular proteins including SAMHD1, and the epigenetic regulator complex HUSH, and likely manipulates innate responses in complex ways. Other studies have used unpurified viral stocks (Manel et al., 2010; Yan et al., 2010), which may contain producer cell proteins, including cytokines, that can activate or stimulate target cell sensing capacity. The method by which primary cells are purified and differentiated may also affect their activation status and hence sensing capacity. For example Decalf et al. describe activation of an IFN response in primary macrophages that was not dependent on reverse transcription or genomic RNA, attributing this response instead to detection of viral entry, dependent on the presence of a viral envelope (Decalf et al., 2017). In this study MDM were prepared by positive selection using CD14⁺ beads, which may activate the cells in ways that other purification methods do not.'*

On reflection we agree with the reviewer's point that this discussion should be factual noting that accurately comparing the dose of virus used between studies is complicated by the varying methods to quantify viruses and the different cell types used. We have therefore removed reference to viral doses used in other studies.

16. In Fig 6, it is unclear if SAMHD1 knocked-down cells were used?

These are monocytic cells without SAMHD1 depletion. We have made this clear in the text (line 411): 'To test this, we infected monocytic THP-1 IFIT-1 reporter cells with increasing doses of HIV-1 GFP' and figure legend.

17. The title is too broad. Most of the data has been performed in THP-1 cells, and one experiment in MDM. This should be made clear in the title.

As described above we have now provided significantly more primary macrophage data and therefore feel that the title is justified.

Referee #3

In this manuscript from Sumner et al provide evidence that the structure of the HIV-1 capsid protects nascent viral reverse transcripts from cytoplasmic DNA sensing. Specifically, they show that viruses produced in the presence of low doses of protease inhibitors, capsid-destablizing drugs, or incorporation of gag proteins with mutations in the CA-SP1 PR cleavage site results in viruses that trigger cGAS/STING dependent interferon responses upon sensing of RT products in myeloid cell lines and primary macrophages.

This is an interesting, well-written and important manuscript. The authors have very carefully controlled their experiments to rule out contaminating DNA and other IFN-inducing impurities in their viral stocks, something that is sadly not the case for many high-profile papers published in this field. Thus, the authors make a compelling case that manipulating Gag processing at levels that have small effects on one-round viral infectivity, leads to exposure of reverse transcripts to cGAS/STING.

I have a few suggestions to enhance the general interest of the manuscript and the interpretation of key results that the authors could consider:

- The implication from much of the data is that low doses of PIs, at around the IC50, can cause the trigger IFN. It would be nice to try and put some of these observations into a more physiologically relevant context - i.e. to provide evidence that the efficacy of protease inhibitors is enhanced by the

triggering of antiviral responses, which is something the authors skirt around without really addressing. Somethings that authors could consider are the following.

o The use predominantly of one-round virus infection throughout the MS, as the authors rightly point out, would not be expected to reveal an antiviral IFN phenotype. However, a spreading infection in human macrophages may do so. The prediction would be that inhibition of replication in macrophages in the presence of low-dose, but not high-dose, PI concentrations would be at least partially interferon dependent.

We have performed spreading infections in primary MDM in the presence of low dose LPV with the IFN blocking drug ruxolitinib, or IFN receptor blockade using an IFN receptor-targeting antibody. These interventions, now included as appendix figure S2, preventing IFN activity, did not rescue infection. This result is expected because inhibition of protease, and the consequent block to infection, cannot be rescued by IFN blockade. Even if IFN is produced, as we have evidenced, the virus remains poorly infectious because it is the PI inhibition of viral protein cleavage that inhibits infection. We have now included these data as appendix figure S2 and included the following text (line 290):

'We also performed similar experiments measuring replication of HIV-1 in MDM over several days, inhibiting replication with various concentrations of LPV. In this case, neither blockade of IFN receptor with antibody, or inhibition of JAK/STAT signalling with ruxolitinib, significantly rescued infection over two independently performed experiments (Appendix Fig. S2A-D). We hypothesise that prevention of IFN activity does not rescue viral replication because the replication inevitably remains suppressed by effective protease inhibition. However, in vivo, we might expect that IFN produced in this way would contribute to innate and adaptive immune suppression of infection.'

Note also that our new data mentioned above examines the effect of IFN blockade after single round infection with PI treated virus. In these new experiments IFN blockade does not rescue infection levels of DRV-treated virus during single round infection of primary MDM (Fig. EV1L), whilst rescue of infection is observed with ruxolitinib during primary MDM infection with the 75% Δ CA-SP1 mutant (Fig EV2E&F). These results are discussed in the following text (line 567):

'There was also no rescue of LPV or DRV inhibited HIV-1 replication in primary MDM with either ruxolitinib or IFN receptor blockade with antibody (Fig. EV1L, Appendix Fig. S2). Conversely infection by the 75% Δ CA-SP1 HIV-1 mutant in primary MDM was rescued by ruxolitinib (Fig. EV2E, F). Thus this virus, made by mixing WT and mutant Gag constructs, causes IFN production that subsequently inhibits viral infection. We expect that primary MDM may exhibit a faster or more potent IFN response allowing IFN mediated suppression of infection even during single round infection of these cells. This result is reminiscent of rescue of infectivity, with IFN receptor blockade, of HIV-1 bearing CA mutants P90A and N74D in MDM (Rasaiyaah et al., 2013). We hypothesise that whether IFN inhibition rescues infection or not depends on the degree to which the suppression of replication depends on IFN secretion. For example, IFN inhibition does not rescue replication in PI-treated MDM because protease inhibition is sufficient for the observed viral inhibition. Conversely, in the case of Gag cleavage mutants (Fig. EV2E, F), single round infection is in part inhibited by IFN and thus, JAK/STAT inhibition rescues some degree of infectivity. In vivo, whether the PI inhibited virus is directly suppressed by IFN or not, we would expect IFN to contribute to innate and adaptive immune responses and globally contribute to viral inhibition.'

Importantly this new data unequivocally demonstrates the antiviral effects of the IFN response induced in MDM during infection with a capsid defective virus.

o PI resistance in PR comes at a fitness cost that leads to second site mutations in the Gag cleavage sites. Do PI resistance mutations in PR trigger more IFN in macrophages/THP-1 and if so to the second-site mutations in Gag rescue it?

This is a very interesting point raised by the reviewer and something that we are currently pursuing in collaboration with Prof Ravi Gupta's lab. Unfortunately the mechanism of protease resistance by changes in Gag is not well defined. Recent unpublished work is further characterising protease

resistance mutations but, unlike resistance mutations in RT, protease resistance mutations are context dependent and diverse. We therefore feel that such an analysis is beyond the scope of the current manuscript.

- The authors suggest that CA structure is compromised, but in reality the 30nM treatment of LPV is sufficient to trigger the majority of the IFN, and this is still fully capable of saturating T5a in FRhK. Thus, these viruses are sufficiently wt in shape to bind TRIM5, implying the compromisation is more subtle. Capsid stability seems an obvious thing to look at - is there any further information the authors can add on this score.

We agree with the reviewer that the defective particles may only be compromised subtly (particularly at 30nM LPV), and this may not easily be revealed by in vitro biochemical assays. We aimed to perform the cGAS-vDNA IP suggested by reviewer 2 to test whether there is a greater interaction between viral DNA and cGAS during infection with the defective viruses to address this point but this has not been possible due to closure of the lab. In general we feel that the data presented in figure 5 demonstrate that PI-treatment and Δ CA-SP1 mutation leads to aberrations in capsid formation and this is supported by published electron microscopy studies (Muller et al, 2009; Schatzl et al, 1991; Mattei et al, 2018).

- WBs for STING, cGAS and MAVS in the KO cells (and CRISPR lesions) should be included.

The STING^{-/-} and MAVS^{-/-} THP-1 cell lines were previously described (PMID: 30061100, line 694 of the manuscript) and immunoblots are provided in this publication. The cGAS^{-/-} cells were purchased from Invivogen (line 651). All cell lines were validated phenotypically by stimulation with DNA, RNA etc... (Fig. 4A and 4B).

- The MAVS observation with DRV is weird. Is it incoming viral gRNA or virion associated RNAs?

The small contribution of MAVS to CXCL-10 production (although not IFIT-1 reporter activity) in DRV-treated virus infected cells is indeed intriguing and suggests some small contribution of RNA sensing to signalling activation. However, this is a small effect and beyond the scope of this current manuscript. We discuss this observation from line 607:

'An interesting finding that warrants further investigation is the observation that MAVS contributed to CXCL-10 production in response to infection with DRV-treated virus (Fig. 4G), but did not contribute to the corresponding IFIT-1 reporter activity (Fig. 4E). MAVS-dependent pathways are known to activate transcription factors other than IRF-3, such as NF- κ B (Seth et al, 2005), which also contributes to the production of CXCL-10 (Yeruva et al, 2008), but not activation of the IFIT-1 reporter (Grandvaux et al, 2002). It is therefore possible that activation of MAVS by HIV-1 contributes to NF- κ B activation in these cells but not an IRF-3 response.'

Dear Rebecca,

Thank you for submitting your manuscript to The EMBO Journal. I am sorry for the delay in getting back to you with a decision, but I have now received the needed input on your revision.

Your manuscript has been seen by the original referee #2 and 3. While referee #3 is happy with the revised version, referee #2 is not convinced that the manuscript is a strong consideration for publication here. I have discussed the remaining concerns further with referee #3 and we find that the raised issues don't preclude publication here and that they can be addressed with suitable text changes and inclusion of the mentioned reference. No further experiments are needed. Let me know if we need to discuss the comments further - happy to do so.

When you submit the revised version will you also please take care of the following issues:

- Please re-label 'Declaration of Interests' as conflict of interest
- The appendix files need to be added to one Appendix file including their figure legends. The appendix file needs a ToC as well.
- Is it possible to combine two of the EV figures into one? We normally can only have 5 EV figures.
- Some of the blots are a bit over-contrasted can you take a look and see what you think.
- We also need a Data Availability statement in the manuscript. As your manuscript doesn't have data that needs to be deposited in a database - please correct me if I am wrong - simply say => Data Availability: This study includes no data deposited in external repositories.
- We encourage the publication of source data, particularly for electrophoretic gels and blots, with the aim of making primary data more accessible and transparent to the reader. It would be great if you could provide me with a PDF file per figure that contains the original, uncropped and unprocessed scans of all or key gels used in the figure? The PDF files should be labeled with the appropriate figure/panel number, and should have molecular weight markers; further annotation could be useful but is not essential. The PDF files will be published online with the article as supplementary "Source Data" files.
- We include a synopsis of the paper (see <http://emboj.embopress.org/>). Please provide me with a general summary statement and 3-5 bullet points that capture the key findings of the paper.
- We also need a summary figure for the synopsis. The size should be 550 wide by [200-400] high (pixels). You can also use something from the figures if that is easier.

That should be all - let me know if we need to discuss anything further

With best wishes

Karin

Karin Dumstrei, PhD

Further information is available in our Guide For Authors:

The revision must be submitted online within 90 days; please click on the link below to submit the revision online before 30th Sep 2020.

Referee #2:

The authors have improved the study with new data and clarifications in the text and figures. The overall model is convincing in THP-1 cancer cells. The data remains weak and the model insufficiently demonstrated in MDM; claims in the abstract and the text are too broad. They could not perform new experiments to address the localization and the interaction between cGAS and the viral DNA as a function of capsid stability. Considering the recent literature on this topic, novelty

appears incremental.

1) The concept of capsid stability being critical for cGAS sensing of the HIV DNA has been recently established, in a 2018 study from the Yamashita lab published in the Journal of Virology (PMID 31167922). It is surprising that this study is not cited. The Yamashita study came to very similar conclusion as the current study, using hyperstable capsid and PF-74. That study also showed the role of cGAS and provided direct evidence for altered capsid stability. Thus, the concept is not novel. Here the novelty comes from the use of several destabilized capsid settings instead of an hyperstable capsid; these findings are interesting but incremental.

2) The added data in MDM (Fig 2I, 2K, 2J) is consistent with the idea that altering CA induces innate immune activation also in these primary cells, but they have not tested the role of cGAS and the viral DNA synthesis in MDM. This remains an important weakness because primary MDM from multiple donors are very different from THP-1 cells cancer cells isolated from one donor, even more so when these THP-1 cells are treated with the NFkB activator PMA, as in many experiments here. It could well be that MDM sense capsid-destabilized HIV-1 through another pathway than DNA/cGAS. In abstract (l.27-29) and text (l.80-81) they must restrict the claims on cGAS and DNA synthesis to THP-1 cells only.

3) They respond that they used THP-1 IFIT1 reporter cells that expressed SAMHD1 in some experiments. This is important because it validates some of the findings in SAMHD1-positive THP-1 cells. However, they also use THP-1 IFIT1 reporter cells with SAMHD1 shRNA in about as many experiments. It is unclear why they keep changing the status of this restriction factor. Since they write the cell type in each figure panel, they should indicate when SAMHD1 has been knocked-down as well. This is important because obviously HIV-1 does not abrogate SAMHD1.

4) They confirm that statistical tests have been performed on technical replicates. They did not aggregate the data from biologically independent experiments in figures. They indicate in their response that the data between experiments varies too much in magnitude, resulting in noisy analysis. In spite of this limitation, they have enough multiple independent lines of evidence to support their model in THP-1 cancer cells. However, the data with primary cells (MDM) remains a very weak part of the study. The data would have been more convincing if it had been reproduced and aggregated on multiple donors. Variability is inherent to the use of primary cells from independent human donors, which is why many donors are very often required to establish biological, and statistical, significance.

Referee #3:

The authors have addressed my points, although I might take issue with the interpretation of figure S2.

The JAK/STAT inhibitor which is probably a much more robust inhibitor of IFN signaling than the Ab is clearly rescuing a proportion of viral replication in the presence of the lower, but not the higher, dose of LPV. This is exactly the phenotype I suggest that the authors might see if they did this experiment. Likewise, they will probably find a significant difference in IC50 for LPV in presence or absence of Rux. I appreciate that given the COVID19 that it is probably unreasonable to expect the authors take this further, but to me it looks promising.

All in all this is an interesting and valuable study and should be published.

Referee #2:

The authors have improved the study with new data and clarifications in the text and figures. The overall model is convincing in THP-1 cancer cells. The data remains weak and the model insufficiently demonstrated in MDM; claims in the abstract and the text are too broad. They could not perform new experiments to address the localization and the interaction between cGAS and the viral DNA as a function of capsid stability. Considering the recent literature on this topic, novelty appears incremental.

1) The concept of capsid stability being critical for cGAS sensing of the HIV DNA has been recently established, in a 2018 study from the Yamashita lab published in the Journal of Virology (PMID 31167922). It is surprising that this study is not cited. The Yamashita study came to very similar conclusion as the current study, using hyperstable capsid and PF-74. That study also showed the role of cGAS and provided direct evidence for altered capsid stability. Thus, the concept is not novel. Here the novelty comes from the use of several destabilized capsid settings instead of an hyperstable capsid; these findings are interesting but incremental.

This study from 2019 is now cited and a sentence has been added to the discussion (lines 394-397) as follows:

'These findings are supported by a recent study demonstrating that HIV-1 bearing a PF-74-resistant, hyperstable capsid mutation (R143A) showed reduced cGAS-dependent innate induction, again linking capsid stability to sensing (Siddiqui et al., 2019).'

The novelty of our study comes from our use of protease inhibitors and Gag cleavage mutants rather than the use of mutants with altered biochemical properties. Together these studies strengthen the model that intact capsids protect viral DNA from cGAS. Our study also brings a new angle that widely used protease inhibitors could activate innate sensing by disturbing capsid function.

2) The added data in MDM (Fig 2I, 2K, 2J) is consistent with the idea that altering CA induces innate immune activation also in these primary cells, but they have not tested the role of cGAS and the viral DNA synthesis in MDM. This remains an important weakness because primary MDM from multiple donors are very different from THP-1 cells cancer cells isolated from one donor, even more so when these THP-1 cells are treated with the NFkB activator PMA, as in many experiments here. It could well be that MDM sense capsid-destabilized HIV-1 through another pathway than DNA/cGAS. In abstract (l.27-29) and text (l.80-81) they must restrict the claims on cGAS and DNA synthesis to THP-1 cells only.

We have changed the abstract (lines 27-29) as follows to address the point raised above by the reviewer:

'Importantly, unlike wild-type HIV-1, infection with cleavage defective HIV-1 triggered an IFN response in THP-1 cells that was dependent on viral DNA and cGAS. An IFN response was also observed in primary human macrophages infected with cleavage defective viruses.'

We have also amended the introduction (line 84) as suggested by the reviewer:

'This response in THP-1 cells was mostly dependent on viral DNA synthesis and the cellular sensors cGAS and STING.'

3) They respond that they used THP-1 IFIT1 reporter cells that expressed SAMHD1 in some experiments. This is important because it validates some of the findings in SAMHD1-positive THP-1 cells. However, they also use THP-1 IFIT1 reporter cells with SAMHD1 shRNA in about as many experiments. It is unclear why they keep changing the status of this restriction factor. Since they write the cell type in each figure panel, they should indicate

when SAMHD1 has been knocked-down as well. This is important because obviously HIV-1 does not abrogate SAMHD1.

Stable SAMHD1-depleted THP-1 cell lines were used in experiments where differentiation with PMA was used, as these cells would otherwise infect very poorly with HIV-1. The figure legends detail where PMA treatment and cells that are depleted for SAMHD1 (labelled 'shSAMHD1') have been used. Where 'shSAMHD1' is not specified in the legend the cells are not depleted for this molecule. We hope this clarifies any confusion.

4) They confirm that statistical tests have been performed on technical replicates. They did not aggregate the data from biologically independent experiments in figures. They indicate in their response that the data between experiments varies too much in magnitude, resulting in noisy analysis. In spite of this limitation, they have enough multiple independent lines of evidence to support their model in THP-1 cancer cells. However, the data with primary cells (MDM) remains a very weak part of the study. The data would have been more convincing if it had been reproduced and aggregated on multiple donors. Variability is inherent to the use of primary cells from independent human donors, which is why many donors are very often required to establish biological, and statistical, significance.

The primary MDM data provided in figures 1K and L are indeed aggregated data from 2 donors as the reviewer is suggesting and statistical significance was obtained. The new primary MDM data in figures 2I-K and EV1K-L are representative of 2 repeats where the same phenotype was observed in both experiments, but the magnitude of innate induction was unfortunately too different to permit aggregation of the data. Repeating these experiments more times was not possible due to COVID19 enforced lab closure. We do not feel however that repeating the experiments further would have changed the conclusions we present in the manuscript.

Referee #3:

The authors have addressed my points, although I might take issue with the interpretation of figure S2.

The JAK/STAT inhibitor which is probably a much more robust inhibitor of IFN signaling than the Ab is clearly rescuing a proportion of viral replication in the presence of the lower, but not the higher, dose of LPV. This is exactly the phenotype I suggest that the authors might see if they did this experiment. Likewise, they will probably find a significant difference in IC50 for LPV in presence or absence of Rux. I appreciate that given the COVID19 that it is probably unreasonable to expect the authors take this further, but to me it looks promising.

All in all this is an interesting and valuable study and should be published.

We thank the reviewer for these interesting suggestions and hope to pursue this further for future publications once the lab reopens.

Dear Rebecca,

Thanks for submitting your revised version. I have now had a chance to take a look at it and all looks good. I am therefore very pleased to accept the manuscript for publication here.

Congratulations on a nice paper!

with best wishes

Karin

Karin Dumstrei, PhD
Senior Editor
The EMBO Journal

Please note that it is EMBO Journal policy for the transcript of the editorial process (containing referee reports and your response letter) to be published as an online supplement to each paper. If you do NOT want this, you will need to inform the Editorial Office via email immediately. More information is available here: http://emboj.embopress.org/about#Transparent_Process

Your manuscript will be processed for publication in the journal by EMBO Press. Manuscripts in the PDF and electronic editions of The EMBO Journal will be copy edited, and you will be provided with page proofs prior to publication. Please note that supplementary information is not included in the proofs.

Should you be planning a Press Release on your article, please get in contact with embojournal@wiley.com as early as possible, in order to coordinate publication and release dates.

If you have any questions, please do not hesitate to call or email the Editorial Office. Thank you for your contribution to The EMBO Journal.

** Click here to be directed to your login page: <http://emboj.msubmit.net>

Corresponding Author Name: Dr Rebecca P Sumner

Manuscript Number: EMBOJ-2019-103958